# Abrupt onset of intensive human occupation 44,000 years ago on the threshold of Sahul

Ceri Shipton [1,2,3] ✉, Mike W. Morley [4] ✉, Shimona Kealy [2,3] ✉,
Kasih Norman [2,5,6], Clara Boulanger [2,3,7,8], Stuart Hawkins [2,3],
Mirani Litster[2,9], Caitlin Withnell[1] & Sue O'Connor [2,3]

Archaeological evidence attests multiple early dispersals of *Homo sapiens* out of Africa, but genetic evidence points to the primacy of a single dispersal 70-40 ka. Laili in Timor-Leste is on the southern dispersal route between Eurasia and Australasia and has the earliest record of human occupation in the eastern Wallacean archipelago. New evidence from the site shows that, unusually in the region, sediment accumulated in the shelter without human occupation, in the window 59–54 ka. This was followed by an abrupt onset of intensive human habitation beginning ~44 ka. The initial occupation is distinctive from overlying layers in the aquatic focus of faunal exploitation, while it has similarities in material culture to other early *Homo sapiens* sites in Wallacea. We suggest that the intensive early occupation at Laili represents a colonisation phase, which may have overwhelmed previous human dispersals in this part of the world.

The dispersal of our species out of Africa presents a disjunct between fossil and genetic evidence. The fossil record indicates repeated *Homo sapiens* presence in Eurasia from Marine Isotope Stage (MIS) 8 (300–243 ka) onwards[1–4], reaching mainland Southeast Asia by MIS5a (82–71 ka)[5] and Sumatra by early MIS4 (~68 ka)[6]. However, genetic evidence indicates that over 90% of non-African ancestry is derived from a single late dispersal during MIS4 (71–57 ka), reaching Sahul (the combined continent of Australia and New Guinea at times of lowered sea level) in MIS3 (57–29 ka)[7–12]. A trace of an earlier dispersal may survive in the genomes of some Papuans[13], while the archaeological site of Madjedbebe shows humans had already reached Australia in late MIS4 (~65 ka)[14]. One possibility is that there were multiple dispersals of our species into Sahul[15], with a major post-MIS4 dispersal overwhelming and obscuring the genetic signature of earlier events. Here we explore this hypothesis through renewed excavation and analysis of Laili, the oldest yet known human occupation site on the eastern side of the Wallacean archipelago that lies between Eurasia and Sahul[16,17].

Timor is the largest of the remote eastern Wallacean islands (Fig. 1). Dispersal modelling shows that Timor would have been a key waypoint on a southern migration route through Wallacea and potentially the jumping-off point to reach Australia[15,18–20]. The island can only be reached from the west by crossing strong ocean current flows, and no fossils nor archaeological remains have been found that might be attributed to any hominin other than *H. sapiens*[17]. Previous attempts to date human dispersal into eastern Wallacea have been hampered by the lack of non-human large terrestrial animals living in caves and rockshelters, which would otherwise introduce organic sediment that would then trap minerogenic sediment[21]. As a result, initial human site occupations typically occur directly on bedrock or

[1]Institute of Archaeology, University College London, London, UK. [2]ARC Centre of Excellence for Australian Biodiversity and Heritage, The Australian National University, Canberra, ACT, Australia. [3]Archaeology and Natural History, College of Asia and the Pacific, Australian National University, Canberra, ACT, Australia. [4]Flinders Microarchaeology Laboratory, Archaeology, College of Humanities, Arts and Social Sciences, Flinders University, Adelaide, SA, Australia. [5]Australian Research Centre for Human Evolution, Griffith University, Griffith, QLD, Australia. [6]Centre for Archaeological Science, School of Earth, Atmospheric and Life Sciences, University of Wollongong, Wollongong, NSW, Australia. [7]Japan Society for the Promotion of Science, Department of Modern Society and Civilization, National Museum of Ethnology, Osaka 565-8511, Japan. [8]UMR 7194 Histoire Naturelle de l'Homme Préhistorique, Muséum National d'Histoire Naturelle, Paris, France. [9]Archaeology, College of Humanities, Arts and Social Sciences, Flinders University, Adelaide, SA, Australia. ✉e-mail: c.shipton@ucl.ac.uk; mike.morley@flinders.edu.au; shimona.kealy@anu.edu.au

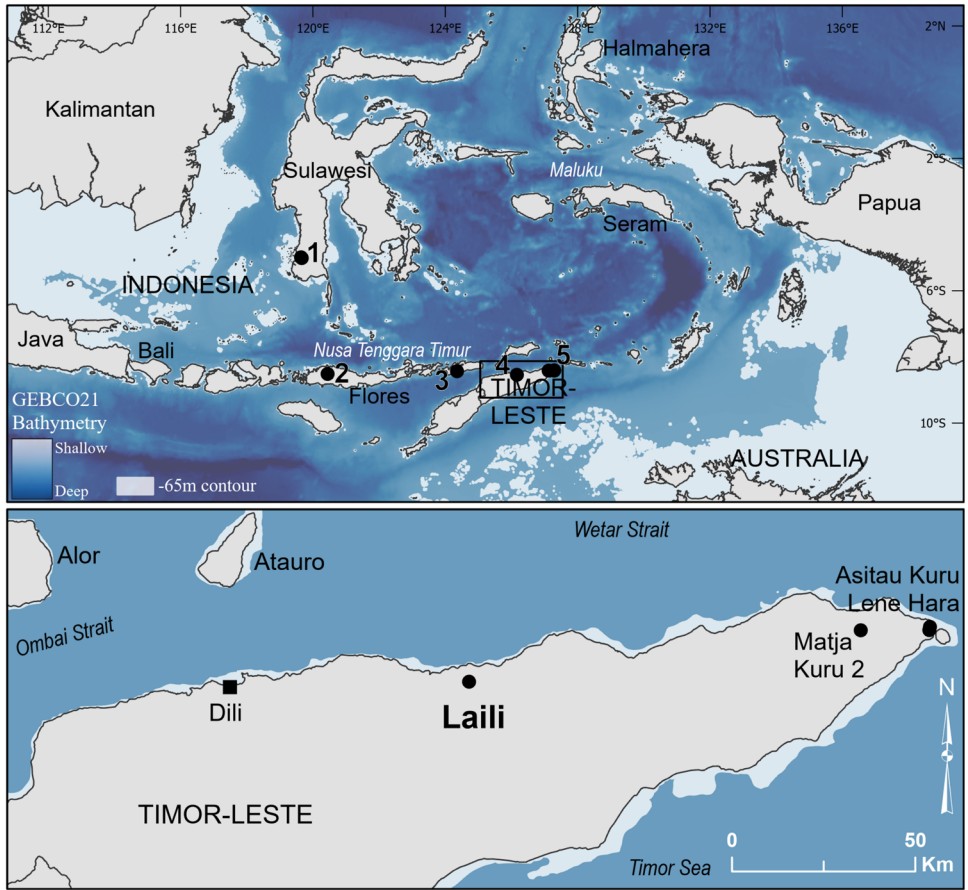

**Fig. 1 | Map of Wallacea and surrounding region (TOP) showing the location of Laili on the north coast of Timor (BOTTOM).** Numbers on the top map correspond to the following archaeological sites mentioned in the text: (1) Leang Burung 2, Leang Bulu Bettue, and Leang Tedongnge; (2) Liang Bua; (3) Makpan and Tron Bon Lei; (4) Laili; (5) Matja Kuru 2, Asitau Kuru, and Lene Hara. The −65 m contour shading reflects the likely land extent at ca. 45 ka[98]. Base map created using the GEBCO21 grid[99].

beach deposit, meaning it is not possible to obtain dates on pre-human deposits for these sites.

Laili is a large rockshelter (~15 × 15 m) in a prominent formation of Carboniferous crinoid limestone boulders overlooking the lower Laleia river and its floodplain on the north coast of Timor (Fig. 1, Supplementary Fig. 1). Today the site is 4.3 km from the coast, with the steep bathymetric profile meaning it would have still been within 5 km during the lower sea levels of MIS3[16]. Laili is unusual as sediment, including organic components, has accumulated there through colluvial and aeolian input rather than human agency. Thus there are basal deposits underlying the main human occupation sequence with the potential for dating. Furthermore, the sedimentary infill is distinctive, well preserved, and extends down to at least 3 m in depth, affording a detailed reconstruction of the geomorphic history of the site.

An initial 1 m² test excavation (Square A) in 2011 at Laili obtained a date of 44 ka (MIS3) from a piece of charcoal (D-AMS-007344) in a hearth immediately overlying sediment apparently lacking evidence for human occupation[16]. In this study we expanded the main excavation to 4 m² in order to characterize the material culture and intensity of this MIS3 occupation at a greater level of detail. We undertake micromorphological (microstratigraphic) analyses of the sediments to address questions regarding the onset and nature of human occupation of the site, as well as obtaining further age estimates for both the cultural and underlying sediments. This Wallacean sedimentary sequence offers insights into Late Pleistocene dispersals of our species.

## Results
### Dating

Twenty layers of human occupation, as well as eight cut features and sixteen fills were identified in the excavation (Fig. 2; Supplementary Fig. 2). These were underlain by an apparently culturally sterile layer whose lower boundary was not reached. Bayesian modelling of 53 radiocarbon and 11 OSL dates (Supplementary Data 1, Supplementary Notes 1–3) indicates the phase of pre-human sedimentation at Laili (Phase 0, Layer 21) ended at approximately 53,733 years ago (median date), with a 95.4% probability range of 63,598–44,990 years ago (Supplementary Note 3). The median date for initial human occupation was 43,949 (Start of Phase 1) with a 95.4% probability range of 48,558–42,803 years ago. The slight overlap in these two probability ranges is supportive of human occupation beginning at the site without a sedimentary hiatus in the record between these two phases. A second MIS3 occupation phase (Layers 18–16) lasts from 37,847–36,237 years ago (Supplementary Note 3). After a hiatus between 36,237 and 26,452 years ago, the Laili deposit continues until the beginning of the Holocene (10,106 years ago), with a final additional phase of mid-late Holocene occupation (9419–1878 years ago) preserved in breccia deposits on the cave walls (Supplementary Note 3).

### Stratigraphy and micromorphology

Laili contains a thick and complex stratigraphic sequence that is strikingly anthropogenic in origin from Layer 20 upwards. Notably, the sediments infilling Laili are exceptionally rich in ashes, to the extent

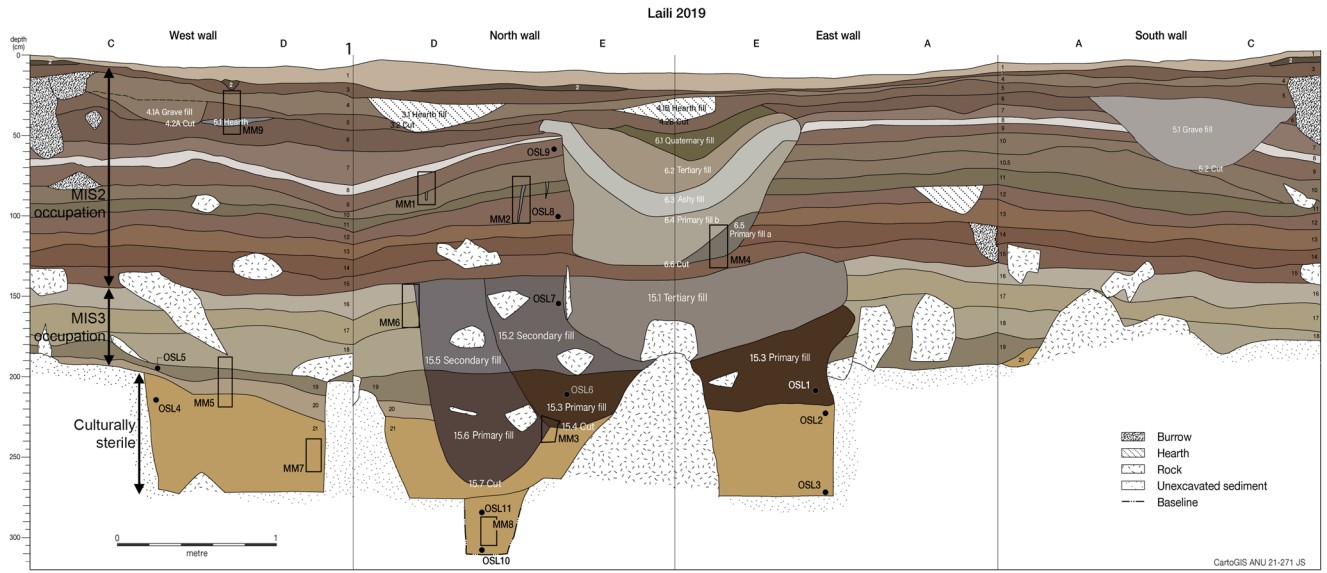

**Fig. 2 | Laili Squares A, C, D, and E excavation profile showing layer numbers, location of OSL and micromorphology samples.** The sondage is visible in the north wall at the base of the excavation.

that the fine sediment matrix is dominated by this material. The Laili stratigraphy also exhibits marked changes in sediment colour and texture from one lithological unit to the next, suggesting sudden shifts in occupation history and/or rapid changes in depositional environment. The occupation deposits we focus on here are summarised below.

Some anthropogenic materials were recovered from a small sondage (Fig. 2) into the lowest excavated portion of Layer 21. However, direct dating of a piece of charcoal and two pieces of marine shell (Supplementary Data 1), as well as intrusive sediment adhering to the lithics (Supplementary Fig. 3), confirmed that these were from a rodent burrow.

Basal Layer 21 is culturally sterile and geogenic in nature, formed of weakly stratified yellowish-brown, calcareous silt with a porous structure (Fig. 3A). Coarse inclusions are common and include frequent elongate shell fragments (Fig. 3B), occasional small pieces of bone, as well as various rock grains including weathered limestone. These silts contain localized organic-rich regions in areas immediately below the Layer 21–20 interface, where darker material infills voids spaces consistent with animal burrowing (Fig. 3A). The bioturbation we observe at the microscale in thin section is not, however, sufficiently large to allow the movement of stone artefacts. The pale-yellow sediments of Layer 21 were once a naturally occurring substrate deposited on the floor of the shelter. They were likely primarily deposited by colluvial and aeolian action. There are no obvious signs of significant pauses in sediment accumulation, with any ephemeral breaks in sedimentation not sufficiently long-lived to result in distinct weathering (pedogenic) horizons.

The Layer 21/20 interface is striking, forming an abrupt and well-defined boundary between the basal silts (Layer 21) and the dark, organic-rich sediments of Layer 20, visible at both micro and macroscale (Fig. 3C, D; Supplementary Fig. 4). Layer 20 comprises dark brown silts and clays with a poorly sorted mix of bone and shell fragments, limestone spalls, organic inclusions, and composite grains ranging from fine silt to coarse sand and fine gravel, with the fine matrix clay-rich in localized areas. In the lower part, elongate inclusions are often horizontally aligned, including shell and rock fragments. Burnt bones, ashes, and charcoal are ubiquitous in this layer and the fine matrix is commonly iron-stained. Clearly there were combustion features very close to the sampled area as we see intact lumps of stratified ash and charcoal.

The characteristics of Layer 20 are consistent with an archaeological occupation horizon of intensive human use of the shelter. The abrupt interface with the silts of Layer 21 suggest that human activity started intensively rather than as a gradual build-up. Microstratigraphic evidence for human presence in the shelter is evidenced by frequent angular fragments of chert debitage related to the production of stone artefacts (Fig. 3E), and frequent combustion byproducts such as ash and charcoal from intensive pyrotechnological activity (Fig. 3F, H, I). Localized horizontally aligned crushed shells indicate trampling on a surface, while the poorly sorted and chaotic composition in other areas suggest dumping and mixing, perhaps as raking out practices in fireside areas. On a macroscale, discrete combustion features were evident horizontally (Supplementary Fig. 5).

Layer 19 marks a shift to finer-grained sedimentation, with an increase in organic matter and finely divided charcoal powder, both of which create a very dark colouration (Fig. 3J, L, M). The coarse fraction heterogeneity and chaotic alignment of the inclusions in the lower part of Layer 19 may suggest that this is a dump of occupational detritus rather than a living surface. Charcoal fragments and flecks are very common and angular chert inclusions and bone fragments are also present (Fig. 3K). Ash is preserved in very localized areas. Some bone fragments are calcined and very pale under transmitted light, indicative of high combustion temperatures driving out the organic component of the bone. The proliferation of combustion byproducts suggests an increase in the use of fire in the rock shelter in comparison with Layer 20 (although we cannot rule out functional changes in site areas). The presence of charcoal with well-preserved cellular structure shows that combustion residues did not move far from their original source, consistent with the macroscale combustion features we observed (Supplementary Fig. 5).

Layer 19 is much darker towards the upper interface with Layer 18, correlating with an increase in clay and fine silt (Fig. 3L). The porosity of Layer 19 decreases here, with horizontal banding, increasing occurrences of horizontal planar voids, and a decrease in sediment porosity, changes which are consistent with trampling and indicate an occupation horizon. A reduction in calcium carbonate suggests decalcification which accords with evidence of chemical diagenesis, such as gypsum-filled voids (Fig. 3M) and reaction rims on bone fragments indicating probable phosphatization. These features present in the dense and organic-rich upper part of Layer 19 are consistent with a living floor at Laili.

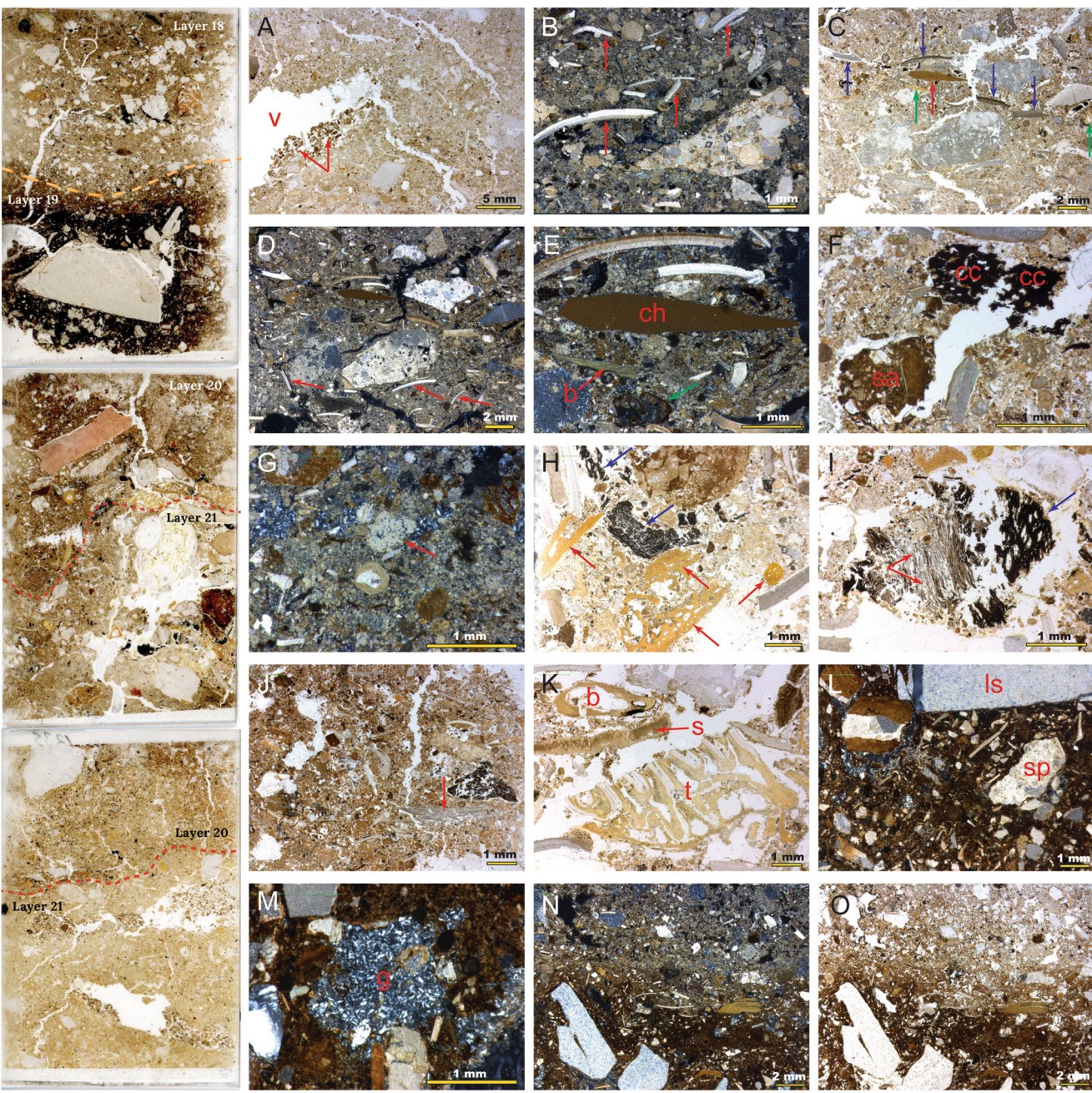

**Fig. 3 | Profile (left) formed of three thin sections through Layers 21–18, and photomicrographs (right) of sediment details. A** Layer 21, yellow silts with occasional crushed shell (probably land snail, Camaenidae) and infilled void spaces (v) with dark silts (red arrows) indicating minor vertical translocation of occupation material from Layer 20; **B** Layer 21, crushed shells (red arrows) showing natural accumulation of molluscs and deposition of coarse pre-occupation colluvium; **C, D** Layer 21/20 interface, with dark, organic sediment and anthropogenic inclusions. Horizontally aligned chert debitage fragment (red arrow) on lower boundary of Layer 20 and coarse shell fragments (blue arrows), bone (green arrow), and rock fragments deposited on trampled surface. In cross polars. **D** Layer 21 is calcareous prior to deposition of organic-rich Layer 20. Haphazard orientation of Layer 21 shell fragments (red arrows) are consistent with colluviation; **E** Layer 20, detail of chert debitage (ch) in cross polars. Sharp bone fragment (b) in a better preservation state than weathered bone from Layer 21. Sediment staining (Fe) is also evident, and a probable seed pod in lower part of image; **F** Layer 20. Charcoal (cc) and ash (micritic calcium carbonate) are preserved, often with plant cellular structure intact. A large rounded soil aggregate (sa) suggests reworking from outside shelter; **G** Localised concentration of ash (red arrow), sometimes partially dissolved; **H** Layer 20: well-preserved charcoal (blue arrows) and bone fragments (red arrows). Internal structure preservation suggests limited movement; **I** Displaced combustion feature fragment with laminated plant structures (red arrows) variably burnt with charcoal on what would have been the top (blue arrow); **J** Layer 19 sediment matrix, becoming increasingly organic-rich with finely divided charcoal powder and a darker colouration. Phytoliths at lower right (red arrow); **K** Layer 19, bone (b), tooth (t), and shell (s) fragments; **L** Layer 19, dense organic clays towards upper interface with Layer 18. Increase in coarse inclusions, including speleothem (sp) and limestone (l) fragments; **M** Gypsum crystals (g) infilling a void space; Layer 19/18 interface, showing abrupt transition from organic clays (Layer 19; **N**) to calcareous sediments (Layer 18; **O**).

The transition to Layer 18 marks another abrupt shift in the intensity of site usage (Fig. 3N, O). The interface between layers 19 and 18 is sharp, contrasting the dark clays of Layer 19 to the mid-brown silts of Layer 18. Layer 18 is a poorly sorted clay silt with frequent charcoal, and fragments of shell and bone, some of the latter being thermally modified. The sediment porosity is highly variable with frequent elongate channel voids. Charcoal is well preserved in this layer as fragments rather than dispersed powder. Some large (~5 mm)

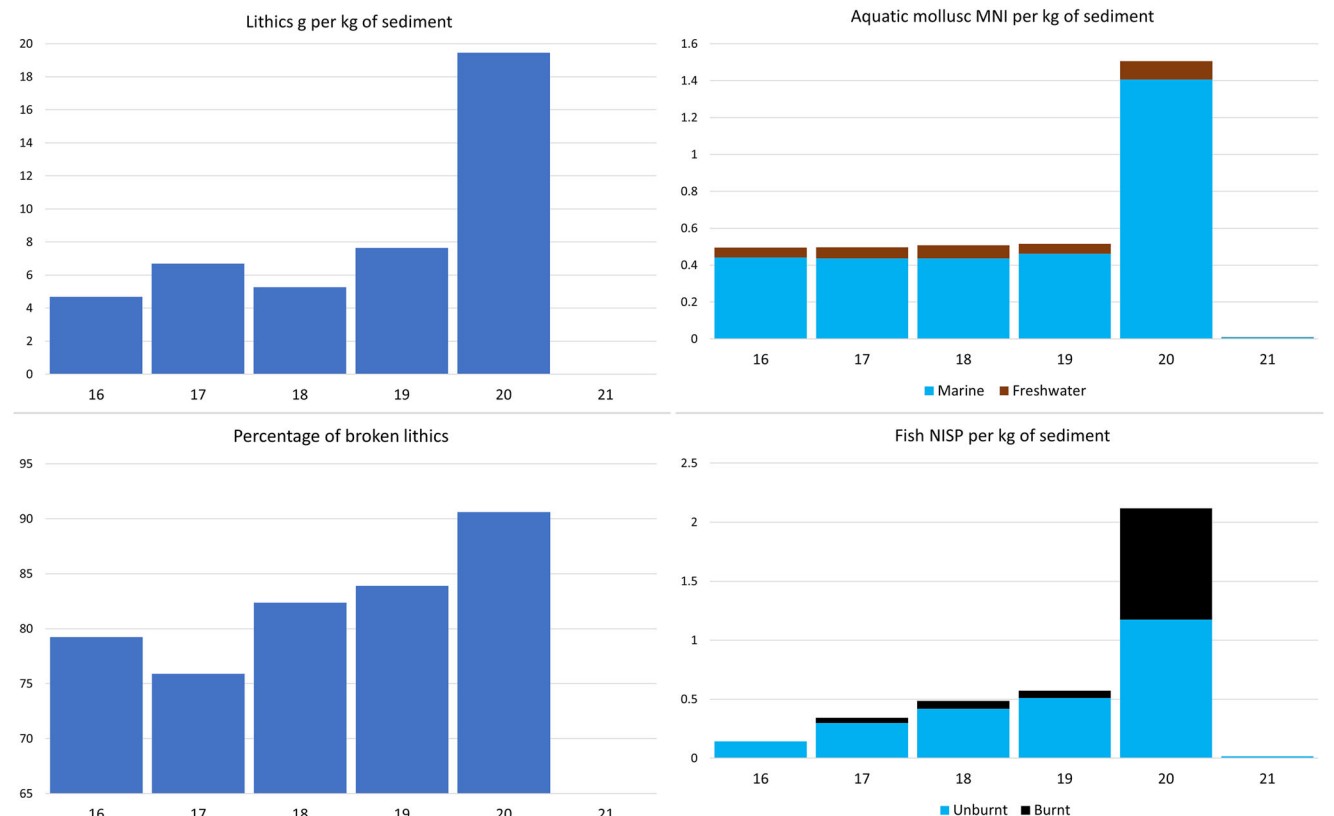

**Fig. 4 | Bar graphs of lithic density and fragmentation rate, and aquatic fauna density by layer for Laili.** Density is per kg sediment for lithics (g), aquatic molluscs (MNI), and fish bone (NISP). Rate of lithic fragmentation is the percentage of broken lithics. Data for lithics and fish cover Squares C, D, and E, data for aquatic molluscs is for Squares D and E only. Source data are provided in the Source Data file.

fragments exhibit plant cellular structure consistent with limited reworking in this part of the stratigraphy. Organic remains are common, often comprising iron-stained vegetal matter and occasional seeds. The increase in plant macrofossils and large charcoal fragments may signify a change in subsistence or could be a preservation bias. Significant quantities of charcoal flecks and powder suspended in the sediment matrix suggests some minor reworking.

A highly calcareous granular silt typifies Layer 17, which has frequent inclusions of small shell fragments, ashes, mineral grains, and clay-silt compound aggregates. There are localized areas of higher porosity with large chamber voids. A speckled fine fabric suggests intensive bioturbation by burrowing animals such as insects, and this is supported by the chaotic arrangement of coarse inclusions, including vertically oriented shell fragments. There appears to be a reduction in anthropogenic features in this layer, although finely divided charcoal is relatively common. We note that the sediments of Layer 17 and Layer 16 above have a very different character and composition than those below (21–18), with a much more open framework matrix.

The interface between Layers 17 and 16 is very diffuse. The sediments contain many weathered features, including decalcified and phosphatised mineral grains and organic material and void spaces that are common around the interface. A defining characteristic of Layer 16 is the increasingly heterogeneous composition with an absence of internal stratigraphy. The lack of stratification in Layer 16 suggests either significant Laili bioturbation or *en masse* deposition of this material, or both. Some large inclusions, including bone, are not only weathered but appear to have been transported over some distance such as a fish scale fragment with clay coating that suggests sub-aerial transport. These sediment dynamics are confirmed by the presence of clay compound grains which are likely pedogenic in origin, having been reworked from areas immediately outside the shelter. This may reflect

an increase in sheetwash colluviation, rapidly moving down sediments from upslope, perhaps related to a wetter and more dynamic environment.

### Flora and fauna
*Celtis* seeds (hackberry) are absent in Layer 21, but are found from Layer 20 upwards, suggesting they were anthropogenically introduced to the sequence (Supplementary Fig. 6). The seeds are more common in Layer 20 than any other MIS3 layer, bar 17.

Fragmentary land snail (Camaenidae) shells occur in Layer 21 (Supplementary Table 1) indicating some natural input. An increase in density of shells, with more complete specimens from Layer 20, and a high density across Layers 19–16 (Supplementary Fig. 7), may suggest human consumption or at least inadvertent introduction by humans on vegetation being brought to the site. Aquatic molluscs are dominated by marine Turbinidae (including *Lunella cinerea*), *Acanthopleura* sp., and *Nerita* (*Nerita exuvia*, *Nerita polita*, *Nerita textilis*), as well as freshwater *Stenomelania* sp. (Supplementary Table 1). They occur in negligible numbers in Layer 21, then are found at much higher frequencies in the main occupation layers, with a pronounced peak in Layer 20 including the highest proportion relative to land snails (Supplementary Table 1) and the highest proportion of marine relative to freshwater species (Fig. 4). In addition to the molluscs, occasional crab and barnacle shells were also found from Layer 20 upwards.

Tetrapod bone recovered from the MIS3 layers of Laili Square C was dominated by murids, mainly small but with some giant specimens (Supplementary Table 2). Other tetrapods were herpetofauna (lizards and snakes) and frogs, as well as microbats and occasional fruit bats (Supplementary Table 2). Some bird bones were found in Layer 21, but these increased in abundance across Layers 20–16 suggesting a growing role as human prey (Supplementary Table 2). The bones that

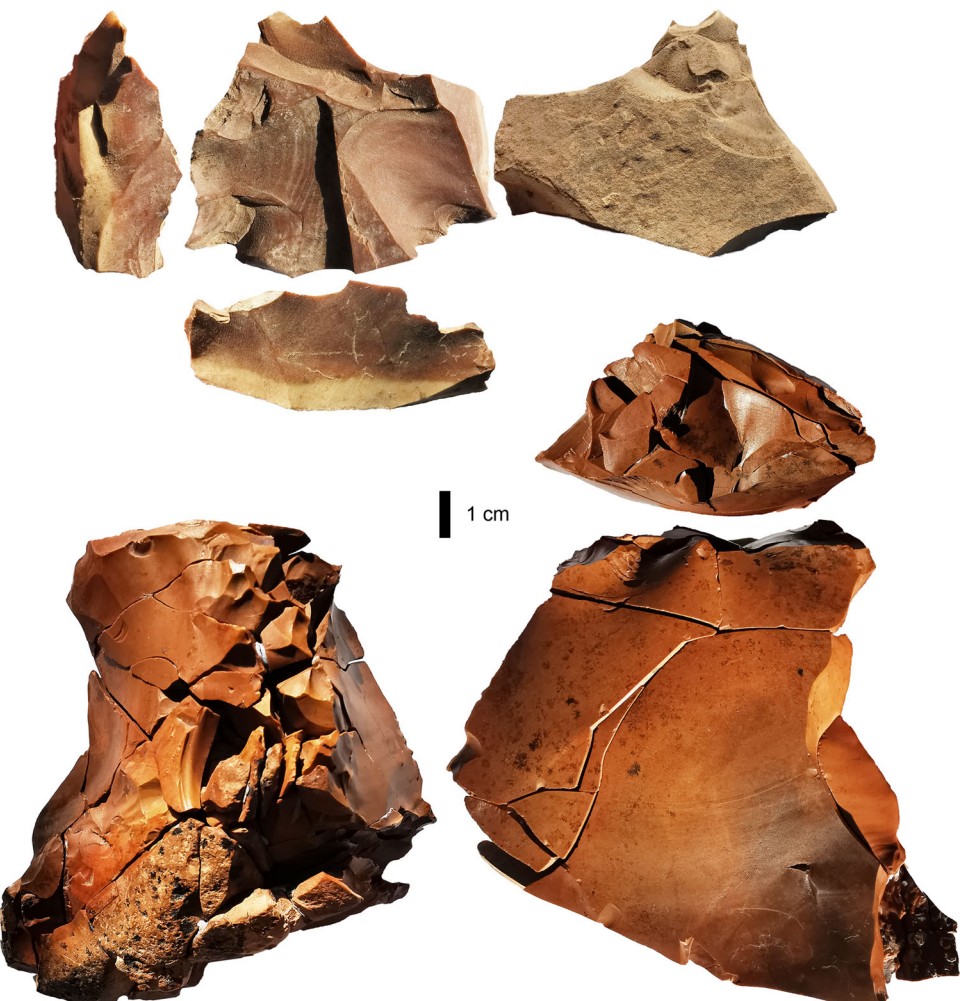

**Fig. 5 | Multifunctional lithics from Laili Layer 20.** Above, hierarchical disc core that was subsequently used as a multiple-notch tool. Below, refitted flake large flake from Laili Layer 20 that was reduced into at least 70 smaller flakes with some of these used as tools.

could be most confidently attributed to human subsistence were fish, which, both in terms of density within sediment and proportion relative to other non-murid tetrapods, were rare in Layer 21, but well-represented across Layers 20–16 (Fig. 4; Supplementary Table 2). Many of the fish bones were burnt, including a particularly high proportion in Layer 20 (Fig. 4). Fish taxa across all squares of Layers 20–16 included freshwater eels, estuarine favouring drums and mullet, as well as marine grouper, needlefish, and trevally[22]. Layer 20 Square C featured the highest proportion of fish among the non-murid tetrapods, the only layer in which they constituted a majority of taxa (Supplementary Table 2). A piece of turtle shell in Layer 19 Square C provides further evidence of aquatic exploitation. Two large varanid teeth were found in Layer 21 Square C but none in the overlying sequence, suggesting an extirpation with the onset of Layer 20.

## Stone artefacts

Knapped stone is abundant at Laili with 6901 pieces recovered from Squares C, D, and E, Layers 16–20. The highest density of lithics in the MIS3 occupation is in Layer 20 (Fig. 4). Over 98% of MIS3 lithics were chert, supplemented by 104 limestone and 17 quartz artefacts. The chert is a high-quality variety with no visible crystalline inclusions and comes in a range of colours with red dominant. It is readily available as large rounded cobbles from the gravels and terraces of the Laleia River, 500 m below the site (Supplementary Fig. 1).

Lithics showed broad consistency in technology across the five layers. Crushed platforms indicative of bipolar technology were evident on 17–22% of complete flakes ($n = 1099$) across layers (Supplementary Table 3), with bipolar cores recovered from Layers 19 and 16. Three pieces were classified as core-on-flakes (in Layers 17, 18, and 19), with resultant Janus flakes comprising 6–16% of dorsal scar patterns across layers (Supplementary Table 3), indicating this strategy was used throughout the MIS3 occupation. Dihedral platforms were present on 4–8% of complete flakes, suggestive of a bifacial core strategy, with examples of discoidal cores (Layers 16 and 18) and a hierarchical disc core (Layer 20) (Fig. 5), as well as four redirecting flakes with old bifacial platforms. Less formal flaking strategies are indicated by seven examples of multi-facial cores in the MIS3 Layers, with a uni-facial core found in Layer 19. Complete flake platform angles, which vary according to technological strategy, were homogenous across the layers (Kruskal–Wallis H = 1.329, $n = 683$, two-tailed $p = 0.856$, median 68°). Overhang removal was commonly used as a platform preparation technique, being evident on 11–19% ($n = 176$) of flakes across layers (Supplementary Fig. 8). Blades ($n = 167$) constitute 15% of the entire complete flake assemblage (Supplementary Fig. 8), but prismatic blade technology was not apparent. One possible limestone hammerstone was recovered from Layer 16, while only 5 bending initiations occurred among the complete flakes suggesting soft-hammer flaking was not a feature of the knapping.

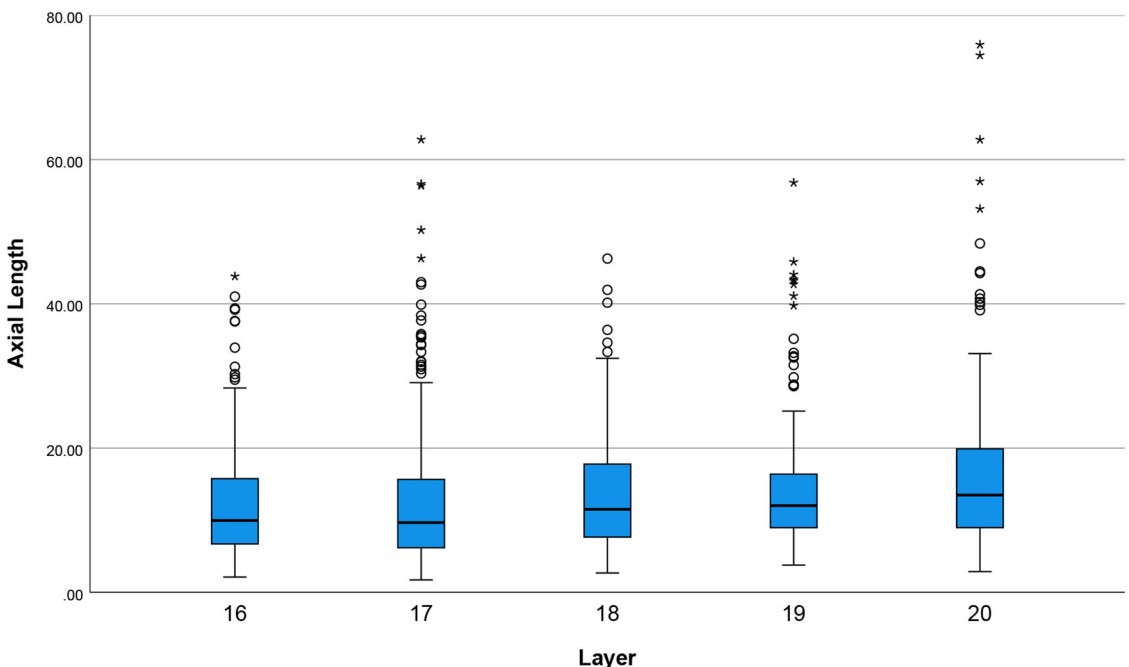

**Fig. 6 | Boxplot of complete flake length by layer for Laili Squares C, D, and E.** Black bars show the median value, boxes the interquartile range, whiskers show 1.5× the interquartile range. Circles represent outliers >1.5–3× the interquartile range, asterisks represent outliers >3× the interquartile range. Layer 20 $n = 171$, Layer 19 $n = 125$, Layer 18 $n = 173$, Layer 17 $n = 209$, and Layer 16 $n = 128$ individual artefacts. Source data are provided in the Source Data file.

Flakes were consistently miniaturized throughout the sequence, with mean flake lengths of <20 mm (Fig. 6). Unusually for a crypto-crystalline material such as chert, initial clast sizes were large (Supplementary Fig. 9) and cobbles were available close to the site, therefore material economy does not explain the small size of flakes produced.

Notches were the most common type of retouched artefact throughout the five layers, comprising over half of all retouched pieces (Supplementary Table 3). A mean notch width of 6.69 mm (SD = 3.5, $n = 30$) suggests these were used in creating small diameter rods, likely of organic material[23].

Initial functional microscopy analysis shows that miniaturized pieces were utilized; such as the scraper from Layer 18 shown in Fig. 7, which measures 12.39 mm in maximum dimension. The pitted micro-wear, invading from the edge on the ventral surface with associated microscars, suggests it was used to scrape a soft material (such as an animal product) perpendicular to the cutting edge[24,25].

Pot lidding (heat damage) is frequent in the assemblage, evident on 10% of all complete lithics, testifying to the intensive nature of the occupation (Supplementary Table 4). Rates of fragmentation were also high, with just 17% of all lithics complete, a feature that was particularly pronounced in Layer 20 where only 9% were complete (Fig. 4). In Layer 20, 10% of complete pieces were retouched flakes, whereas the proportion of retouched artefacts in the other layers declined from 5% in Layer 19 to 1% in Layer 16 (Supplementary Table 4).

A 70-piece refit sequence from Layer 20 supports some of the patterns identified in the technological analysis of complete lithics (Fig. 5). At 13.63 mm, median unretouched final flake size in the refit sequence is equivalent to the miniaturized levels of the main flake assemblage (Fig. 6). The refit sequence, which itself began as a large core-on-flake, includes three second-generation core-on-flakes, and three third-generation core-on-flakes. It also features bipolar knapping, the use of multiple platforms on a single core, the removal of blades, and retouch to create narrow (<10 mm) notches. The functional plurality of the original flake in this refit sequence accords with the hierarchical disc core in Layer 20, as diacritical analysis indicates

the latter was used as a multiple-notch tool in the later stages of its life history (Fig. 5, Supplementary Fig. 10).

A total of 41.9 g of haematite was found distributed across the five layers, including 14 crayons - pieces with macroscopically visible striations (Fig. 8). In addition, a grindstone with extensive ochre residue was recovered from Layer 19 (Supplementary Fig. 11).

## Discussion

Laili is unusual in eastern Wallacea in the decoupling of sedimentation from human occupation. This allows for more accurate timing of human arrival by dating the time when humans were not there in addition to when they were. Furthermore, pre-existing sediment provides a stable surface for initial occupation to accumulate on, allowing it to be characterized as a distinct phase from subsequent occupation.

The first phase of sedimentation at Laili occurred through natural colluvial and aeolian deposition without apparent anthropogenic input. Murids and land snails dominated the site fauna at this time, with some herpetofauna and frogs, as well as birds and occasional varanids. There are no lithics or *Celtis* seeds (other than in large burrows), and the very few marine fauna are likely either intrusive or from bird predation[16]. At the microscale, there are no traces of human occupation such as combustion byproducts or chert debitage. With the Bayesian modelled OSL dates for the excavated upper metre of this layer, we can conclude that humans did not occupy the rockshelter between 59 and 54 thousand years ago.

Human presence at Laili begins abruptly with Layer 20, with the nature of the interface and differences between Layers 21 and 20 suggesting occupation was very intensive from the outset. The dark sediments of Layer 20 are characterized by organic inputs and combustion signatures, including ash, charcoal, and burnt bone. Layer 20 has the highest proportion of burnt fish bone and heat-damaged lithics of any MIS3 layer at the site. Both at the micro- and macro-scale there are abundant chert fragments, and Layer 20 has the highest density of chert lithics by weight of any of the MIS3 layers. Such abrupt and intensive occupation accords with the highest rates of lithic breakage, with the horizontally aligned crushed shell fragments in the

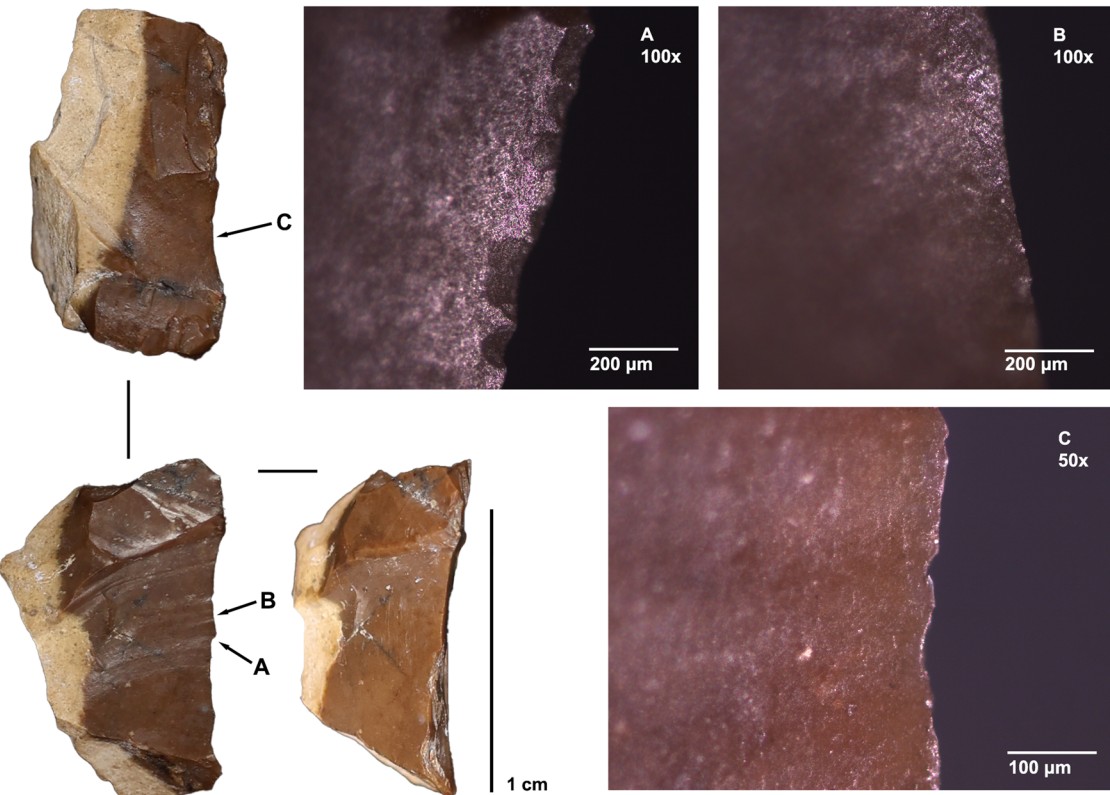

**Fig. 7 | Intermediate polish formation on artefact #923, a scraper from Layer 18.** **A**, **B** show the working edge of the tool on the contact face, with microscars perpendicular to the edge and micro polish that invades inward (100x). **C** shows the opposite (non-contact) face, where Polish development is limited to the immediate edge.

microstratigraphy suggesting trampling on a stable surface. The in situ nature of this initial occupation is further testified by the 70-piece lithic refit sequence, indicating knapping activity on a stable floor surface, with these lithics rapidly buried and preserved by further anthropogenic sediment input from this intensive occupation. Another distinguishing feature of Layer 20 with respect to both the archaeologically sterile Layer 21 and the MIS3 occupation sequence above, is the abundance of aquatic resources. Layer 20 has over three times the density of aquatic molluscs and fishbone as the next densest layers, it has over twice as many aquatic molluscs relative to land snails as any other layer, and it is the only layer to have a majority of fishbone among the non-murid tetrapods.

The evidence for intensive occupation continues with Layer 19 but with a greater frequency of combustion signatures at the microscale, especially towards the upper boundary, and a lower density of lithics at the macroscale. Faunal remains indicate a shift away from aquatic resources with lower densities of fish and marine molluscs and a higher proportion of birds as well as a higher density of land snail. The latter might also reflect or have been prompted by a wetter, more densely vegetated environment. Layer 18 represents a new sedimentation regime, perhaps with a slower deposition rate leading to the destruction of discrete occupation horizons by bioturbation. There is a relatively low density of lithics in this layer and a similar terrestrially oriented resource profile to both Layer 19 and the layers above. Layers 17 and 16 have a lower density of lithics like Layer 18, as well as lower rates of lithic breakage, suggesting less intensive human use of the shelter, allowing for the increased bioturbation observed at the microscale.

The Laili rock formation is prominent in the landscape. The occurrence of freshwater fish bone[22], freshwater molluscs, chert flakes with rounded cobble cortex, in conjunction with bathymetric reconstruction and stable isotopes of chiton shell from Square A[16], suggests the large river transporting chert cobbles would have been active throughout MIS3. The site would thus have provided an optimal location, such that any hominin groups moving through this landscape would likely have made use of the shelter. Therefore, the ages for the archaeologically sterile Layer 21 suggest that human presence on Timor is improbable during the period ~59–54 ka. This aligns with other records of MIS3 occupation from Timor, at the sites of Asitau Kuru and Matja Kuru 2 (Fig. 1), both of which lack records older than 45 ka[26,27].

The onset of intensive occupation at Laili occurs abruptly on an archaeologically sterile surface. Comparison of Laili Square A with other MIS3 occupations in the region (Makpan, Asitau Kuru, Matja Kuru 2) shows that the Laili early occupation can be constrained to a shorter time period and has a higher density of artefacts[28]. That the MIS3 occupation at Laili is particularly intensive at the outset (i.e., in Layer 20) suggests a large-scale colonization; as opposed to an accidental colonization by a small population where we would expect a distinct pattern of slowly increasing anthropogenic material against the backdrop of natural sedimentation. A large migration scenario resonates with broader regional modelling of human colonization that point to planned open-sea voyages across current flows through Wallacea, and founding populations on Sahul >1000 [29,30]. Given the technological repertoire of the early occupants of Sahul, which included string but lacked the adzes essential for dugout canoe production, it is likely that such colonization was on rafts[17,31]. The exploitation of pelagic tuna at Asitau Kuru provides circumstantial evidence in support of such watercraft in the early occupation of Timor[32,33].

Higher proportions of retouch together with artefacts that have complex life histories is consistent with technological strategies for higher mobility in Layer 20 relative to the layers above. This initial

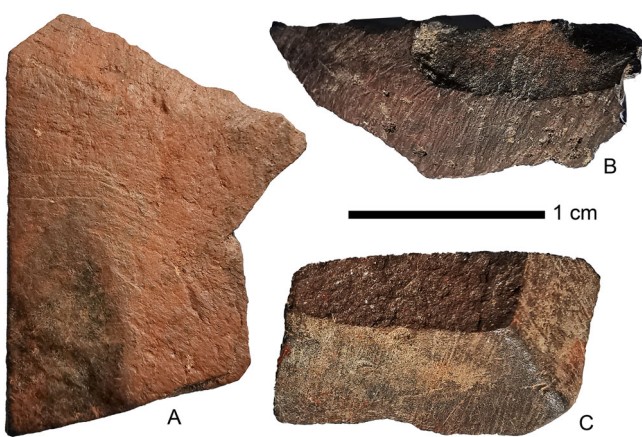

**Fig. 8 | Ochre crayons from the MIS3 layers of Laili. A** and **B** are from Layer Layer 20, **C** is from from Layer 17.

occupation is also distinguished from its successors by the higher proportion of aquatic fauna, which accords with a pioneer phase of island colonization: Aquatic protein sources are more consistent between islands than terrestrial fauna, so would be a more reliable subsistence base for hominins moving from one island to another. This pattern is supported by stable isotope evidence for human diet in the region, with a ~41 ka tooth from the early occupation at Asitau Kuru having a marine diet signature, while later teeth from the same site and others on Timor show more reliance on terrestrial resources[34].

The 44 ka date for the beginning of human occupation in Layer 20 at Laili accords with the MIS3 initial occupation dates elsewhere on Timor of 44 ka at Asitau Kuru and 40 ka at Matja Kuru 2. The final *Homo floresiensis* occupation at Liang Bua on Flores (Fig. 1) dates to 50 ka and the earliest *Homo sapiens* occupation occurs 47 ka[35,36]. Since Liang Bua and Laili are the only two sites thus far known on the Nusa Tenggara Timur (Lesser Sunda) island chain (Fig. 1) which have sediment pre-dating our species' arrival, it is possible that the onset of *H. sapiens* occupation at these sites reflects a dispersal through the archipelago more widely. A transition from an archaic stone stool industry to one associated with *H. sapiens* has been dated to 40 ka at Leang Bulu Bettue on Sulawesi (Fig. 1)[37], perhaps showing the same broad dispersal. Several red-painted rock art images from nearby caves have been dated to at least 40 ka[38,39], the oldest at Leang Tedongnge with a minimum of age 46 ka[40]. A unifying behaviour of the early occupation at Laili and other MIS3 sites on Timor (Asitau Kuru, Lene Hara, Matja Kuru 2), as well as Makpan on Alor (Fig. 1), is the habitual use of red ochre pigment[26,41,42]. Red ochre is associated with early *H. sapiens* more generally, including in Sahul[14,43] and its appearance in archaeological sites in Nusa Tenggara Timur at the same time as its use in paintings in Sulawesi may be a signature of a widespread dispersal of our species.

A specific factor linking the early *H. sapiens* occupation at Laili with that of Matja Kuru 2 and Asitau Kuru at the eastern end of Timor, Makpan on Alor, Liang Bua on Flores, as well as Leang Burung 2 and Leang Bulu Bettue on Sulawesi, is lithic miniaturization[42,44]. Early *H. sapiens* across Wallacea were making particularly small stone tools on cryptocrystalline materials such as chert, using the same range of knapping methods including bipolar and core-on-flake. Even though *H. floresiensis* was using similar methods of knapping in this region[45], their stone tools were not miniaturized[42]. Small flake production by *H. sapiens* in Wallacea is consistent over a broad range of reduction intensity, indicating such items were the goal of knappers[42]. Laili adds a further line of evidence that this lithic miniaturization is a distinctive cultural signature rather than an incidental phenomenon, as it eliminates small clast size and material scarcity as explanations. The locally available chert cobbles and the long refit sequence show small flakes were being produced even on the large clasts at Laili. Furthermore, our

functional microscopy analysis indicates that these small flakes were not merely waste products, but were being utilized.

Lithic miniaturization may even link Laili to sites beyond Wallacea. On Java at the southeastern edge of Sunda, small flakes on siliceous stones have been identified as a key distinguishing feature of *H. sapiens* from MIS3 onwards[46]. In the Kimberley, at the northwestern edge of Sahul, the site of Widgingarri has miniaturized lithics throughout its occupation sequence which begins in early MIS3[47].

The colonization of Europe by our species and the replacement of the incumbent Neanderthal population in MIS3 has been linked to the higher population density of Aurignacian *H. sapiens*[48]. A parallel situation may have occurred at the eastern end of the hominin-occupied world. The Laili sequence shows that a relatively high-intensity occupation began 44 ka at this site along the southern route across Wallacea; with technological and foraging adaptations to facilitate maritime colonization. Rather than a small founding population, this suggests large-scale dispersal[49]. This accords with evidence from both modelling and archaeology for rapid human expansion across neighbouring Sahul in the window ~50–40 ka[47,50]. In contrast, the MIS4 dispersal into Sahul is only known thus far from Arnhem Land where it is best represented by the site of Madjedbebe. This early dispersal may thus have been overwhelmed by a more intensive dispersal, as represented by the initial MIS3 occupation of Laili.

## Methods

### Permissions and legacy
The excavation and export of finds were carried out with permission of the Timor-Leste Ministry of Higher Education, Science and Culture, under permit numbers: 71- and 136- /DGAC-SEAC/MESCC/VI2019, issued on 5th June 2019. Members of the Ministry of Higher Education, Science and Culture participated in the excavation, while a team of Laleia villagers were employed for the duration of the fieldwork to assist with the sorting of excavated material. A poster in Tetum was presented to the Laleia village summarising the results of the previous excavation at Laili, with a fluent Tetum speaker (Lurdes Pires) part of the field team to explain the project. The site is now protected by the Chef de Suco of Laleia from the sediment harvesting that removed the Holocene occupation layers[51].

### Excavation
The 3 m² trench was excavated by stratigraphic layers differentiated according to colour, compaction, and texture, with thicker layers divided into 10 cm spits. Large burrows from rats were emptied and the sediment within them discarded. Anthropogenic cut and fill features were excavated by single context[52], with several large pits identified that had multiple fills (Fig. 2). The Harris Matrix showing the stratigraphic relationship between contexts, Supplementary Fig. 10, was created in Imagination Computer Services Harris Matrix Composer v.2.

The three excavated square metres were adjacent to the previous excavation, Square A, and were labelled Squares C, D, and E (Square B was another test excavation elsewhere in the site), and all material was kept separate between these squares. Due to large rocks reducing the size of the excavation area, the excavation Squares were combined into a single unit labelled CDE from context 37 onwards. Context depths (5 points per square metre) and sediment descriptions were recorded on *pro forma* sheets. Lithics and bone longer than 2.5 cm, and marine shell and charcoal samples for dating were 3D plotted with a total station. Rocks longer than 20 cm were also plotted, as were cut features.

All excavated material was weighed then wet sieved through a 1.5 mm² mesh, before being dried and sorted for smaller stone artefacts, bone, shell, seeds, and charcoal. The remaining rocks and heavy residue were weighed and discarded. *Celtis* seeds were weighed and retained.

## Radiocarbon dating

Thirty-nine radiocarbon samples were sent for radiocarbon age determination from the 2019 excavation season. Charcoal samples were typically collected from hearths identified during excavation and piece-plotted with a total station. Large pieces were collected from pit fills and piece-plotted, with large pieces from two of the upper fills (E6B and GC1) collected from the sieve. One sample from the sondage at the base of Layer 21 was collected during excavation without piece-plotting. Large fragments of marine shell were both piece-plotted and collected during excavation as well as collected from the sieve. Two marine shells from the sondage at the base of Layer 21 were collected during excavation without piece-plotting. Marine shell included both burnt and unburnt specimens. Collection details on individual samples are shown in Supplementary Data 1.

Most charcoal samples were analysed at the Australian National University (ANU) Radiocarbon Centre, while three charcoal samples from the lower layers and all the shell samples were analysed at the University of Waikato Radiocarbon Dating Laboratory. Pretreatment methods for individual samples are shown in Supplementary Data 1.

Pretreatment of charcoal at the ANU Radiocarbon Centre[53] consisted of physical cleaning with a scalpel followed by Acid-Base-Acid (ABA) chemical treatment: The samples were washed in hot HCl, rinsed and treated with multiple hot NaOH washes, then the NaOH insoluble fraction was treated with hot HCl, filtered, rinsed, and dried. After each treatment samples were rinsed in ultrapure water at least three times or until the water remains colourless. Fifteen of twenty-nine samples processed at ANU failed pretreatment.

At the University of Waikato, shell samples were physically pretreated by having their surfaces cleaned with a handheld drill and then being washed in an ultrasonic bath. They were tested for aragonite to calcite recrystallisation through Feigl staining and only aragonite or burnt calcitic samples were used. They were then chemically pretreated using a 0.1 N HCl acid wash, rinsed and dried. The Waikato charcoal samples were washed in hot 10% HCl rinsed and dried, except for one sample from Layer 17 which underwent ABA pretreatment.

## OSL dating

Optical dating provides a means of determining burial ages for sediments—and by association the artefacts and fossils encased within them—by determining the last time quartz or feldspar mineral grains in the sediment matrix were exposed to sunlight[54–56]. For quartz minerals the luminescence 'clock' is reset by just a few seconds of exposure to sunlight. The method is based on the time-dependent increase in the number of trapped electrons induced in quartz mineral grains by low levels of ionising radiation from the decay of natural uranium, thorium, and potassium in the surrounding deposits, and from cosmic rays. The time elapsed since the light-sensitive electron traps were emptied can be determined from measurements of the optically stimulated luminescence (OSL) signals from quartz from which the equivalent dose ($D_e$) is estimated, together with determinations of the radioactivity of the sample and the material surrounding it to a distance of ~30 cm (the environmental dose rate). The $D_e$ divided by the environmental dose rate gives the burial time of the grains (and associated cultural material) in calendar years ago.

Eleven samples were collected for optical dating from depths of between 50 and 305 cm below surface (Fig. 2; Supplementary Table 5). Sediment samples for optical dating were collected from the North, East and West exposed profiles. Samples L19-1–L19-3 were collected from the East profile, samples L19-4 and L19-5 from the West profile, and samples L19-6–L19-11 from the North profile. Sampling was focused on profiles in the northeast and northwest squares of the extended 2 × 2 m excavation to avoid the original test-pit (southeast square), and substantial rockfall (southwest square). All samples were collected using 5 cm diameter sample tubes. Samples L19-1 to L19-3, L19-10 and L19-11 were collected in 20 cm length plastic tubes, and L19-

6 to L19-9 were collected in 15 cm stainless steel tubes, with choice dependent on the compactness and rock content of sediments down profile. Two side-by-side sample tubes were collected for both L19-10 and L19-11, the deepest samples. Samples were sealed immediately after collection. Following collection of the sediment for optical dating a field gamma spectrometer (FGS) probe of either 1-inch or 2-inch diameter was inserted at each sample location to measure the gamma dose rates directly.

Sample field and lab codes, together with mid-point depths of the sample tubes below surface are provided in Supplementary Table 5. As approximately two meters of Holocene-age sediments have been removed from the site in the recent past, both depths below present surface, and estimated depth below original Holocene surface are provided. Samples are presented in order of increasing depth from surface. Some samples were collected from pit-fill features that relate to later depositional events than other samples at equivalent or shallower depths (Fig. 2; Supplementary Fig. 2).

Laili optical dating samples were prepared using standard procedures. All sediment samples were opened under red light laboratory conditions. Samples were then put through a series of nested sieves to sort them into size fractions. The 180–212 μm and 90–125 μm diameter fractions were treated with hydrochloric (HCl) acid to remove carbonates, followed by a hydrogen peroxide ($H_2O_2$) solution to destroy organic matter. Heavy minerals and feldspar minerals were removed from the samples using sodium polytungstate solutions of 2.7 g/cm$^3$ and 2.62 g/cm$^3$, respectively. Following this the quartz mineral grain component of the samples were etched for 45 min in 40% hydrofluoric (HF) acid to remove the alpha-irradiated outer surface layer of the quartz grains. An HCl acid rinse was used to remove any precipitated fluorides from the HF-etched quartz grains and the samples were sieved again. Grains of 90–125 μm diameter were used for optical dating because there were inadequate amounts of 180–212 μm grains.

OSL measurements of quartz grains were made for all samples on an automated Risø TL-DA-20 luminescence reader equipped with a focused green (532 nm) laser for single-grain stimulation. Single grains of quartz were individually transferred from the 90–125 μm sieve onto aluminium discs drilled with 100 holes 300 μm in diameter and 300 μm deep[57]. Grains were moved carefully across the disc surface and into each hole under magnification using a fine brush trimmed back to a few hairs. Any extra grains were removed with the point of a pin that had been rubbed to produce static electricity. Irradiations were carried out inside each luminescence reader using $^{90}Sr/^{90}Y$ beta sources that have been calibrated using a range of known gamma-irradiated quartz. Spatial variations in beta dose rate to individual grain positions were taken into account for $D_e$ determination[58]. We considered reader-specific counting statistics as sources of error and used an instrumental irreproducibility value of 2%[59].

Single-grain quartz measurements on quartz were made using the single-aliquot regenerative dose (SAR) procedure[60,61] with experimental steps listed in Supplementary Table 6.

A total of 16,900 individual quartz grains were measured (between 500 and 3600 grains per sample) representing many months of laboratory work. Grains with aberrant luminescence characteristics rendering them unsuitable for $D_e$ determination were identified using the following standard criteria[62,63]:

1. Initial $T_n$ signal is less than 3σ above the corresponding background count[62], or the relative error on $T_n$ is >25%[64].
2. Recuperation ratio (i.e., the ratio of the $L_x/T_x$ values for the 0 Gy and maximum regenerative doses) is >5%.
3. Recycling ratio (i.e., the ratio of $L_x/T_x$ values for the duplicate regenerative doses) is inconsistent with unity at 2σ.
4. OSL IR depletion ratio is less than 2σ below unity.
5. $L_x/T_x$ ratios are too scattered to be reliably fitted with a curve, or have a large figure-of-merit (FOM) value with an upper limit of 10%, or a have a reduced chi-square value of >5.

6. $D_e$ value is obtained by extrapolation of the fitted DRC, rather than interpolation among the regenerative-dose signals.
7. $L_n/T_n$ ratio is statistically consistent with, or higher than, the saturation level of the corresponding DRC, so that a finite $D_e$ value and error estimate could not be obtained.

Rejection of grains based on the above criteria was accomplished using the function calSARED()in the R-package numOSL[65,66]. Supplementary Table 7 lists the numbers of individual grains measured, rejected, and accepted for $D_e$ determination for each of the samples, and the reasons for grain rejection. There were much higher rates of acceptance in the occupation layers, likely due to the increased sensitivity of these grains from having been heated by the combustion features in these layers of the site (see main text as well as Supplementary Fig. 5). All accepted grains provide reliable estimates of $D_e$. Also listed is the number of accepted grains that have negative $D_e$ values consistent with being modern (zero $D_e$) at 2σ.

Dose recovery tests on single grains from sample L19-2 were used to (a) select the preheat temperatures, and (b) to evaluate the laboratory performance of the single-grain OSL procedure[60,67]. The grains were exposed to natural sunlight for two days to evict electrons from light-sensitive traps and reset the luminescence signal to zero. The grains were then given a laboratory beta dose of 490 s to act as a surrogate 'natural' dose. The grains were then measured using the SAR procedure (Supplementary Table 6) using three different PH combinations to establish if the surrogate natural dose could be recovered. One thousand grains were measured for each preheat combination: (1) 220 °C for 10 s ($PH_1$) and 160 °C for 5 s ($PH_2$), (2) 260 °C for 10 s ($PH_1$) and 160 °C for 5 s ($PH_2$), and (3) 260 °C for 10 s ($PH_1$) and 220 °C for 5 s ($PH_2$). All three sets of measurements resulted in a weighted mean dose recovery ratio (i.e., the ratio of measured dose to given dose) consistent with unity at 2σ. The 260/160 °C combination yielded the ratio closest to unity (0.96 ± 0.06) and with the smallest OD (2 ± 1%). This combination was selected to measure the $D_e$ values for all samples. The dose recovery results for the three data sets are shown in Supplementary Table 8.

OSL signals were estimated from the first 0.22 s of decay, with the mean count recorded over the last 0.3 s subtracted as background. Sensitivity-corrected dose-response curves were then constructed using a general-order kinetic (GOK) function[68], and the sensitivity-corrected natural OSL signal from each grain was projected onto the corresponding fitted DRC to estimate the $D_e$ value. All data analyses, including curve fitting, $D^e$ determination and error estimations, were achieved using the functions implemented in the R-package 'numOSL'[65,66] and *Luminescence*[69].

Supplementary Fig. 11 shows a selection of representative OSL decay curves and corresponding dose-response curves for eight grains of quartz for one sample from the upper profile (L19-8) and one sample from near the bottom of the profile (L19-11A). The OSL decay curves exhibit a range of shapes, but are generally consistent and decay rapidly to instrumental background, typically reaching background within the first ~0.5 s. The majority have very similar shapes and continue to grow with an increase in dose up to 80–120 Gy, after which some begin to saturate while others continue to grow to higher doses.

$D_e$ values from quartz grains were obtained for all eleven samples. The $D_e$ distributions are shown as radial plots[60] (Supplementary Fig. 12). The single-grain $D_e$ values from Layers 9, 12, and 19, and from the primary and secondary fills of the pits have $D_e$ values that are tightly clustered, with the majority within ± 2 standardised estimates of a common value (Supplementary Table 9, Supplementary Fig. 12). The overdispersion (OD) values range between 27 ± 2% (L19-9) and 48 ± 2 % (L19-5) and are typical of samples that have been well-bleached prior to deposition and have remained largely undisturbed since burial, the primary assumptions in optical dating. While the sample from Layer 19 (L19-5) has the largest OD value (48 ± 2 %), >95% of the 580 grains have

$D_e$ values that are consistent with the deposit having remained intact since burial. All samples contain a small number of 'outlier' grains with both smaller and larger $D_e$ values compared to the main component. Outlier grains may either be intrusive grains from younger and older deposits, or may result from beta heterogeneity in the sampled sediment[70].

The single-grain $D_e$ values for samples from Layer 21 (the deepest stratigraphic layer) have OD values between 67 ± 4% (L19-10A) and 138 ± 13% (L19-4) (Supplementary Table 9). These values are higher than expected for samples that were well-bleached prior to deposition and have remained undisturbed since burial. All samples from this lowest layer have a population of grains with much smaller $D_e$ values compared to the main component (Supplementary Fig. 12). Two of the samples also have grains with negative $D_e$ values (Supplementary Table 7). The comparatively higher OD for the samples from Layer 21, suggest the presence of a secondary component interpreted as intrusive from the younger overlying deposits, with the presence of 'modern' grains indicating that there was some post-depositional mixing of the Layer 21 sediment.

To estimate the sample $D_e$ values, the central age model (CAM)[60] was applied to the single-grain $D_e$ distributions after rejection of statistical outliers and grains with negative $D_e$ values. The CAM assumes that the $D_e$ values for all grains are centred on some average value of $D_e$ (similar to the median) and the estimated standard error takes account of any $D_e$ overdispersion. Statistical outlier grains were identified using the normalised median absolute deviation (nMAD) method[71,72]. Grains with log $D_e$ values with nMADs of greater than 2.5 were excluded from calculation of the final $D_e$ value for age determination, and are shown as open triangles in the radial plots (Supplementary Fig. 12). The remaining grains that represent the true burial age of the sediment deposit are shown as filled circles in the radial plots.

The most overdispersed sample was one of two taken close to the overlying Layer 20 from the five Layer 21 samples, L19-4. The finite mixture model (FMM)[73] was applied to the single-grain $D_e$ distribution of L19-4 to determine the number of discrete $D_e$ components, the relative proportion of grains in each component, and the weighted mean $D_e$ value and associated standard error of each component. The minimum number of statistically supported $D_e$ components was estimated using maximum log-likelihood and the Bayes Information Criterion (BIC)[74]. The dominant $D_e$ component of the sample i.e., the component containing the largest proportion of grains (74%) were used for final $D_e$ and age determination and produced similar results to the other sample from the top of Layer 21, L19-2. The remaining ~26% of $D_e$ values in the L19-4 distribution are split between two lower $D_e$ components of ~5 Gy (9%) and ~0.5 Gy (17%).

The total environmental dose rate consists of contributions from beta, gamma and cosmic radiation external to the grains, plus a small alpha dose rate due to the radioactive decay of uranium and thorium inclusions inside sand-sized grains of quartz. To calculate the OSL ages, we have assumed that the present-day radionuclide activities and dose rates have prevailed throughout the period of sample burial.

We estimated the beta dose rates directly by low-level beta counting of dried, homogenised, and powdered sediment samples in the laboratory, using a Risø GM-25-5 multi-counter system[75]. We prepared and measured samples, analysed the resulting data, and calculated the beta dose rates and their uncertainties following the procedures described and tested in Jacobs and Roberts[76]; three sub-samples were measured for each sample. For all samples, allowance was made for the effect of sample moisture content[77], grain size[78], and HF acid etching[79] on beta dose attenuation.

Gamma dose rates were measured directly by in situ gamma spectrometry to take into account any spatial heterogeneity in the gamma radiation field within ~30 cm of each sample (as gamma rays can penetrate this distance through most sediments and rocks). The gamma dose rate was measured at every sample location. Counts were

collected for 1 h with either a 1-inch or 2-inch NaI(Tl) detector. The detectors were calibrated using the concrete blocks at Oxford University[80] and the gamma dose rates were determined using the 'threshold' technique[81]. This approach gives an estimate of the combined dose rate from gamma-ray emitters in the U and Th decay chains and from 40 K.

The cosmic-ray dose rates were calculated[82], and included adjustments for site altitude, geomagnetic latitude, thickness and density of shelter rock and sediment overburden, and the angular distribution of cosmic rays[83].

We assumed an effective internal alpha dose rate of $0.03 \pm 0.01$ Gy/ka. Current water contents of ~10–30% were measured (Supplementary Table 9). We assumed this to represent the long-term water content (i.e., averaged over the entire period of sample burial) with uncertainties sufficient to accommodate the likely range of water contents experienced by these deposits.

Optical age estimates for all samples are listed in Supplementary Table 9. Uncertainties on the ages are given at 1σ (the standard error on the mean) and were estimated by combining, in quadrature, all known and estimated sources of random and systematic error.

## Chronostratigraphic model of Laili

A total of 53 samples (40 charcoal and 13 marine shell) from Laili have been successfully dated by radiocarbon analysis, 22 new to this study and 23 first reported elsewhere[16,51]. F14C values are reported for the new radiocarbon dates in Supplementary Data 1. The additional 11 OSL age estimates described above were obtained from sediment samples collected from the 2019 excavation at Laili. The Bayesian chronostratigraphic model was constructed in OxCal v4.4[84], as a multi-phase model whereby each stratigraphic Layer was modelled as a *Phase* in which the measured ages are presented in stratigraphic order but treated as uniformly distributed by the model. Each Layer *Phase* was ordered by stratigraphic position within the model *Sequence*, thereby assuming that underlying Layers are older than those above. The model code is provided as the file Supplementary Code 1.

Single transition *Boundaries* were placed between each Layer, and double (start and end) *Boundaries* placed where dating of Layers was discontinuous (e.g., between Layer 15 and 13) and/or where a possible temporal gap was identified between two layers. These broader groupings of more continuous age-depth chronostratigraphic associations were identified as Phases 0–7. The use of double *Boundaries* between each Phase maximises model flexibility, allowing for possible variability across the sequence (e.g., changing sedimentation rates, gaps in the record) which is a common occurrence in archaeological assemblages where neither deposition nor dating is likely to be continuous (unlike lake records for example)[85].

Within the model, the radiocarbon dates were calibrated using the Marine20 curve for shell samples[86], and a mixed U(0,50) curve, combining the IntCal20[87] and SHCal20[88] curves, as recommended for terrestrial (i.e., charcoal) samples from the Inter-Tropical Convergence Zone[88,89]. Each OSL age was input as a C_Date in calendar years before 1950, with an associated 1σ error. For charcoal samples, we applied the *Charcoal Plus* t-type Outlier Model with a prior outlier probability of 10%, which is specifically designed to account for the inbuilt age of charcoal (i.e. the old wood effect), while also allowing for some stratigraphic movement in an archaeological context[90,91]. The *General* t-type Outlier Model with a prior outlier probability of 5% was used for the marine shell samples and OSL ages, following commonly used modelling procedures for archaeological dates[85,90]. Six radiocarbon samples (S-ANU 63824, S-ANU 63813, S-ANU 63811, S-ANU 63816, S-ANU 66317, S-ANU 66318) and three OSL samples (OSL_L19-7, OSL_L19-6, OSL_L19-1) associated with pit-fill deposits were assigned an increased outlier probability of 20% to account for the greater uncertainty surrounding their stratigraphic position. Two charcoal samples (D-AMS 001653, D-AMS 001661) and one OSL sample (OSL_L19-9) were

identified as having a greater risk of stratigraphic error due to the context of their recovery (e.g. near a burrow, hearth cut, or other disturbance) and their prior outlier probability increased to 50% to reflect this. In addition, nine radiocarbon samples (Wk_49810, S-ANU 63818, Wk-49811, S-ANU 66319, D-AMS 001660, D-AMS 001663, D-AMS 001662, S-ANU 63821, S-ANU 63820) were identified as outliers based on their unreliable stratigraphic context and/or significant inversion in their dates. These nine outliers were thus excluded from the chronostratigraphic model, although they are still presented alongside our results.

The results of the model are presented in Supplementary Fig. 13, with Supplementary Data 1 containing the complete dating results, the model structure and results, and details on outliers.

## Micromorphology

Micromorphology is the microscopic examination of sediment blocks extracted from excavation profiles[92–94]. The original geometric arrangements within the blocks are retained, allowing for observations to be made of the original physical relationships at a microscopic scale. Micromorphology sediment samples were collected from key regions of the stratigraphy, targeting interfaces between layers. Blocks were extracted by scoring around the area, before extracting with a knife and covering the block in gypsum plaster to preserve structural integrity. Sample blocks typically measure around $20 \times 10 \times 10$ cm. Blocks were shipped to the Microarchaeology Laboratory at Flinders University, where they were oven dried at 35 °C. The dried blocks were impregnated with Dalchem crystic polyester resin diluted with styrene (ratio of 7:4) and catalysed with methyl ethyl ketone peroxide (12.5 ml per litre of resin/styrene mixture). After curing, the samples were oven-dried overnight at 50 °C and trimmed to $50 \times 75$ mm 'wafers'. Thin section manufacture was undertaken at Adelaide Petrographics. Thin sections were first scanned on a flatbed scanner at 9600 dpi, both in reflection mode and without the flatbed cover to provide an overview of the general composition. Thin section examination was carried out with stereoscopic and petrographic microscopes at magnifications ranging from ×8 to ×200 under plane- and cross-polarised light. Thin section terminology follows that of Stoops et al. [95].

## Faunal analyses

Mollusc shells from excavation Squares D and E were identified to the lowest taxonomic level possible. The number of individual specimens (NISP) was counted and weighed by taxonomic group. The minimum number of individuals (MNI) was counted using the most commonly represented non-repetitive element: opercula for *Turbo* sp., the spire for other marine gastropods, and the aperture for land snails.

Tetrapod skeletal elements from Square C were identified to the lowest taxonomic level possible through comparison with Australian National University collections. Taxonomically identifiable fish bones from all three squares were assigned to the lowest level possible through comparison with collections from the Muséum National d'Histoire Naturelle (UMR7209), Paris. Each specimen with evidence of burning was also noted in the case of fish bone.

## Stone artefact analyses

Data collection was conducted in Microsoft Excel v.365. Haematite pieces (red ochre) were weighed and the presence of striations noted. Grinding stones were weighed and inspected for red ochre residue. Knapped stone artefacts (lithics) were classified according to the three main rock types; chert, limestone, and quartz, then were counted and weighed.

Lithic technological types and measurements were conducted in accordance with standard regional practice e.g., refs. 26,96,97. Complete flakes, retouched flakes, and cores were assigned a type. For

complete flakes and retouched flakes their axial (box) length, medial width, and platform angle was measured and their dorsal scars were counted. Platforms and dorsal scar patterns were assigned a type, and the presence of any cortex, platform preparation, heat damage, or macroscopic use-wear was noted. In the case of notched pieces, the width of individual notches was also measured. Cores were oriented according to the axis of the largest complete flake scar, then their axial length, width, and thickness, as well as the length and platform angle of this scar, were measured, and the total number of flake scars counted. Figure 4 and Supplementary Figs. 6 and 7 were created in Microsoft Excel; Fig. 5 was created in IBM SPSS v.29.

Refitting was attempted on 150 distinctive orange coloured chert flakes which occurred in a cluster in Square C Layer 20. A diacritical scar ordering analysis was conducted on a particularly complex lithic from the Laili MIS3 assemblage, a hierarchical disc core that was also used as a multiple-notch. Supplementary Fig. 10 was created in Imagination Computer Services Harris Matrix Composer v.2.

For the microscopy analysis, after macroscopic assessment, a subsample of pieces were washed with a mild solution (2%) of detergent and distilled water in an ultrasonic bath at 25 °C for ten minutes per artefact. Each piece was then rinsed with distilled water to remove any remaining detergent and air-dried overnight on paper towel. If needed, adherent residues were removed using alcohol-impregnated wipes. During and after cleaning, artefacts were handled with gloves to avoid contamination. Stereomicroscopy was performed using a Leica S8 APO B (10–80×, camera: GXCam HiChrome HR4 Hi-Res, software: GX Capture-T), and metallographic microscopy using a Leica DM4500 (50–500×, objectives 5×/NA = 0.12/WD = 14.0 mm, 10×/NA = 0.25/WD = 17.6 mm, 20×/NA = 0.40/WD = 1.15 mm, 50×/NA = 0.75/WD = 1.50 mm, camera: Leica DFC290 HD, software: Leica Application Suite) fixed with a polarised filter and external lighting array. Artefacts were mounted on a glass slide using museum putty for stability during microscopy.

### Reproducibility

Micromorphology features that are diagnostic of specific environmental conditions or human activities were observed in thin section using the point-counting method. 200 points were counted in each thin section and the percent by volume figure calculated for that feature. Where feature counts were ≥10% across particular micro-facies, these were used for the interpretation and a microphoto was selected to best represent this feature. We verified the intra-observer reliability of the data by re-counting areas of selected thin sections.

Lithics with retouch or macroscopic indicators of use, including edge damage, abrasion, and striations were observed with metallographic microscopy, and where microwear features were present on the artefact surface they were recorded with a micrograph, with up to 30 images taken per artefact. These were assessed for diagnostic features and a representative micrograph was chosen. Visualisation of the microwear features was confirmed by repeating the location and identification of the feature before imaging.

### Reporting summary

Further information on research design is available in the Nature Portfolio Reporting Summary linked to this article.

## Data availability

All data used in this study are available in the Supplementary Information, the Supplementary Data file, or the Source Data file. Finds were labelled with the trench code LA19, the square (D/C/E), and the excavation unit (1-44). Finds are temporarily housed at the Australian National University and University College London, but long-term curation will be at the Timor-Leste Museum and Cultural Centre. Source data are provided with this paper.

## Code availability

The OxCal Bayesian model code used in this paper is provided as the Supplementary Code 1.txt file.

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

## Acknowledgements

Funding for this project was provided by the Australian Research Council Centre of Excellence for Australian Biodiversity and Heritage grant awarded to S.O.C. (CE170100015), with additional contributions from a Future Fellowship (FT180100309) awarded to M.W.M. We are grateful to Manuel Ximenes Smith, Diretor Geral da Arte e Cultura, and Gaspar Jose Fatima da Costa, Chefo do Suco de Laleia, for permission to conduct the research. We thank Lurdes Pires, Karene Chambers, Tierney Lu, Marc Verhoeven, the staff of the cultural heritage office, and the villagers of Laili for assistance in the field. We appreciate the guidance of Zenobia Jacobs and Lee Arnold with the OSL dating.

## Author contributions

Conceptualisation: S.O.C., C.S., S.K. Data collection: C.S., K.N., M.W.M., C.B., S.H., M.L., S.K., S.O.C., C.W. Analysis: C.S., M.W.M., S.K., K.N., C.W. Writing: C.S., M.W.M., K.N., with contributions from all authors.

## Competing interests

The authors declare no competing interests.
