## [Peer Review File · Nature Communications]

Abrupt onset of intensive human occupation 44,000 years ago on the threshold of SahulREVIEWER COMMENTS

Reviewer #1 (Remarks to the Author):

This paper provides an interesting perspective on the abrupt human colonization on the eastern part of Wallacea and just at the boundary of Sahul. One of the implications would be on the discussion and critique of the routes of migration to Sahul, considering the genetic and archaeological evidence. The synthesis of this paper warrants it to be published with few minor revisions. These are my general comments:

1. The site is unusual, and the detection of pre-modern human deposits also provide insights into prehistoric migrations through the eastern part of Wallacea. What are the implications of this in the question of modern human migration across Wallacea, including the northern migration route?
2. Could you provide a short paragraph on the means of colonization, sea crossings?
3. Would it be possible to explain further the entry regarding the "optimal location next to a large river with abundant chert"? Is this assumption just based on the current geological pattern in the area or also on the prehistoric record indication production of stone tools? Are these documented already?
4. Is miniaturization an indication of the sizes of the raw material sources rather than an intentional reduction sequence? Also is this more affected by the quality rather than the question of raw material scarcity? Was there an actual survey at least to document the quality of the chert in the area, or at least some experimental flint knapping activity?
5. Regarding the timing mentioned in the text, the timing indicates abrupt and overwhelming migration, however do you address and correlate this with the major migration routes previously proposed?
6. Regarding the stone tools, since we are talking of modification in relation to production and probably use/function, is there any possibility for lithic use-wear analysis? Would be interesting to have an idea of the activities through a functional study, which would also include experimental work on chert tools.
7. The site is unusual in terms of pinpointing the period when humans were not and suddenly intensively occupying the site. Are their indications of the practice associated with "seafaring"?
8. Would it be possible to include a description or illustration of the boundary of the occupation phases mentioned on the stratigraphy?

Reviewer #2 (Remarks to the Author):

This paper reports dating and other results for an archeological site in Timor that may be an example of a migration wave of anatomically modern humans. The report is fairly straightforward, although I have some questions listed in the detailed comments. The authors seem to think this site represents a planned colonization. I do not see any evidence for that. I think the site merely represents the expansion of a human group that may have been better adapted than earlier groups. Humans can build up a debris fairly rapidly. I would not call the site an intense occupation. The higher density artifacts than usual may just represent good preservation in a restricted area.

1. line 30 – I would leave out the word "planned". It implies a degree of intentionality that the authors have no proof of.
2. lines 35-54 – I am not sure I understand the contradiction. The authors propose earlier dispersals

- overwhelmed by a later one. That seems consistent with both the genetic and fossil record.
3. lines 60-61 – Why does it take non-human animals to produce sedimentation in rock shelters? Couldn't the wind do that?
 4. line 70 – If Laili is unusual in having non-human sedimentation, why is that so? What makes it so different from other places.
 5. lines 90-2 – What distinguishes the layers and how are cut and fill features defined? I notice later the layers are distinguished by color and texture. You should say that here.
 6. lines 93-94 – I notice the OSL samples are shown in Figure 2, why not the radiocarbon dates?
 7. lines 177-182 – I think this discussion needs expanding. Why is this layer more of a “living floor” than the other layers. I do not follow the argument.
 8. lines 211-219 – How is en masse deposition consistent with colluvial sheetwash?
 9. line 299 – The fact you could refit 70 pieces together seems remarkable. Is this consistent with an intense occupation, which might scatter the pieces more?
 10. lines 373-378 – The sudden appearance of humans at Laili and other sites in the region does suggest a migration but not necessarily a planned colonization. This paragraph seems unnecessarily speculative and probably should be left out unless the argument can be fleshed out more.
 11. lines 662-713 – I have no idea what I am looking at in these micromorph pictures. For example, in K, I see neither the bone nor the tooth. I wonder if these would work better as separate figures with separate captions, and maybe some arrows to point out particular features. Also N and O seem like the same section but with different imaging. What is the purpose of this? The transition the authors talk about is certainly more visible in O than N.

Supplement

12. lines 9-11 – The caption needs better explanation. I do not see cave breccia in the figure.
13. Figure S5 – I do not see the horizontal combustion features
14. line 143 – Why were 90-125 μm grains used instead of the 180-212? While neither size provides single-grain resolution with the 300 μm holes in the disks, much fewer grains will fit into the holes for 180-212. So you actually have small aliquot not single-grain resolution. This is particularly the case given the high acceptance rates of some of the samples.
15. lines 178-180 – I think more justification is needed for applying this criterion given that a large number of grains were rejected for this reason
16. line 227 – Not sure what you mean by “reproducible” here
17. lines 244-255 – You do not mention the possibility of beta heterogeneity for the higher over-dispersion in these samples.
18. lines 263-264 – I am not sure about applying nMad for removing outliers. How do you know they do not represent a separate component, as some of the radial graphs seem to suggest?
19. lines 269-277 – What value was input for the over-dispersion typical of a single-aged distribution in the finite mixture model? Can you justify using the most abundant component for age determination? Why was the FMM not applied to L19-10A, which had very high over-dispersion?

Reviewer #3 (Remarks to the Author):

[Please see attached pdf documents]

Reviewer #4 (Remarks to the Author):

This is a well put together paper describing the archaeology at Laili, a highly significant site in Timor-Leste, the dating of which has significant implications for the settlement of the area. The timing of human arrival into Sahul is something that has been hotly contested in recent times, so this new information from Wallacea is of great importance. A clear signature of human habitation beginning ~44ka is consistent with dates from the very few other early sites in Timor Leste reported thus far.

The conclusions are well-supported by the archaeological data which is presented, including secure archaeological dates and well-described differences in the material material and archaeozoological deposits in the various chronological layers of the site, clearly showing evidence for changing human use of the site over time (and lack there-of in the case of natural deposits). The site described is rare in the region, with a record going back to first human arrival (not many cave sites in Timor Leste), so reporting of this site and its dates is of great importance.

Standard archaeological conventions have been followed. The analyses and presentation of this data are up to standard.

RESPONSE TO REVIEWERS: *Reviewer comments in italics* **our responses in bold**

Reviewer #1 (Remarks to the Author):

This paper provides an interesting perspective on the abrupt human colonization on the eastern part of Wallacea and just at the boundary of Sahul. One of the implications would be on the discussion and critique of the routes of migration to Sahul, considering the genetic and archaeological evidence. The synthesis of this paper warrants it to be published with few minor revisions.

We thank the reviewer for their positive remarks.

These are my general comments:

1. The site is unusual, and the detection of pre-modern human deposits also provide insights into prehistoric migrations through the eastern part of Wallacea. What are the implications of this in the question of modern human migration across Wallacea, including the northern migration route?

We have added a sentence to the introduction to clarify that Timor is on the southern route (lines 55-57). This paper doesn't really have a direct bearing on the northern route.

2. Could you provide a short paragraph on the means of colonization, sea crossings?

We have added two sentences to the discussion on the probable use of rafts (lines 408-413).

3. Would it be possible to explain further the entry regarding the “optimal location next to a large river with abundant chert”? Is this assumption just based on the current geological pattern in the area or also on the prehistoric record indication production of stone tools? Are these documented already?

We have added two sentences to the discussion on river flow and availability of cobbles (lines 385-391).

4. Is miniaturization an indication of the sizes of the raw material sources rather than an intentional reduction sequence? Also is this more affected by the quality rather than the question of raw material scarcity? Was there an actual survey at least to document the quality of the chert in the area, or at least some experimental flint knapping activity?

We have added sentences addressing this to the stone artefact section (line 270-273). In fact the original paper on Laili (Hawkins et al. 2017) emphasized the high quality and abundance of the chert and suggested it might have been used profligately rather than parsimoniously.

5. Regarding the timing mentioned in the text, the timing indicates abrupt and overwhelming migration, however do you address and correlate this with the major migration routes previously proposed?

We have added a clause to the discussion (line 472-473).

6. Regarding the stone tools, since we are talking of modification in relation to production and probably use/function, is there any possibility for lithic use-wear analysis? Would be

interesting to have an idea of the activities through a functional study, which would also include experimental work on chert tools.

A use-wear PhD is currently working on the stone tools, but we have added her as a co-author and included a new figure (Fig. 7) to show that miniaturized tools are not merely waste products but were actually being used. A paper on the refit sequence is planned but it is too much to include in this paper.

7. The site is unusual in terms of pinpointing the period when humans were not and suddenly intensively occupying the site. Are their indications of the practice associated with “seafaring”?

We have added a sentence to the discussion on pelagic fishing at nearby Asitau Kuru (lines 411-413).

8. Would it be possible to include a description or illustration of the boundary of the occupation phases mentioned on the stratigraphy?

Yes, these have now been added.

Reviewer #2 (Remarks to the Author):

This paper reports dating and other results for an archeological site in Timor that may be an example of a migration wave of anatomically modern humans. The report is fairly straightforward, although I have some questions listed in the detailed comments.

The authors seem to think this site represents a planned colonization. I do not see any evidence for that. I think the site merely represents the expansion of a human group that may have been better adapted than earlier groups. Humans can build up a debris fairly rapidly. I would not call the site an intense occupation. The higher density artifacts than usual may just represent good preservation in a restricted area.

We thank the reviewer for these comments. Not all strata at the site are indicative of intense occupation, but layer 20 is. We now cite a paper (Shipton et al. 2023) showing that this occupation is intensive in comparison to other early sites on Timor.

1. line 30 – I would leave out the word “planned”. It implies a degree of intentionality that the authors have no proof of.

Deleted.

2. lines 35-54 – I am not sure I understand the contradiction. The authors propose earlier dispersals overwhelmed by a later one. That seems consistent with both the genetic and fossil record.

We are pleased that the reviewer is in agreement with the dispersal model presented, we have now replaced the word ‘contradiction’ with ‘disjunct’.

3. lines 60-61 – Why does it take non-human animals to produce sedimentation in rock shelters? Couldn’t the wind do that?

When animals, including humans, occupy caves then sediment accumulates more quickly owing to input of organic material and then ability to trap other sediment follows. There

is a dedicated paper on this topic which we cite: Louys et al. 2017. We have rephrased this and added an additional sentence (lines 60-66), but the point still stands.

4. line 70 – *If Laili is unusual in having non-human sedimentation, why is that so? What makes it so different from other places.*

This is a good question. We have now added an extra clause in parentheses indicating it was likely colluvial and aeolian input, as our geomorphological analyses below indicated (lines 71-73).

5. lines 90-2 – *What distinguishes the layers and how are cut and fill features defined? I notice later the layers are distinguished by color and texture. You should say that here.*

The differentiation of layers, cut, and fill features by colour, texture, compaction, and inclusions is the basis of all archaeology and does not need to be specified here. We have added an extra clause to the methods saying this was by colour, compaction, and texture. Note also that we cite the Museum of London excavation manual in the excavation method, considered the gold standard for excavation method in the U.K.

6. lines 93-94 – *I notice the OSL samples are shown in Figure 2, why not the radiocarbon dates?*

Radiocarbon dates were taken from in situ hearths (considered the most reliable method of sampling) not taken from the section after excavation.

7. lines 177-182 – *I think this discussion needs expanding. Why is this layer more of a “living floor” than the other layers. I do not follow the argument.*

The very dense organic rich sediments at the top of layer 19 include features that are consistent with human activity in that area of the site, rather than a spread of occupational detritus. The horizontally-aligned void spaces suggest trampling occurred on this surface. In layer 20, while we see an abrupt change to anthropogenic accumulation, the occupation/activities may not be occurring in the sampled area. In the upper part of layer 19 we believe what we are seeing is direct human activity on that surface in that area. We have made some minor changes to this paragraph for clarity (lines 179-188).

8. lines 211-219 – *How is en masse deposition consistent with colluvial sheetwash?*

In this sense we mean that sediments were washed rapidly downslope, often as a dump of fine-grained sediments. We have rephrased this for clarity (lines 218-227).

9. line 299 – *The fact you could refit 70 pieces together seems remarkable. Is this consistent with an intense occupation, which might scatter the pieces more?*

Yes, it is remarkable. As we state on lines 352-356 it testifies to the pristine nature of this early occupation which must have been rapidly buried and preserved. It is the rapid burial that is indicative of intensive occupation – we have added a clause to make this point (lines 363-365).

10. lines 373-378 – *The sudden appearance of humans at Laili and other sites in the region does suggest a migration but not necessarily a planned colonization. This paragraph seems unnecessarily speculative and probably should be left out unless the argument can be fleshed out more.*

We agree that this is speculative, so we have added a new paragraph to expand the argument (lines 396-408). We explicitly state the alternative scenario of a gradual build-up in intensity. Then we cite a new study we have recently published showing the early occupation at Laili is more tightly constrained temporally and more intensive in terms of number of lithics than other early Wallacean sites. We also draw a parallel with Wairau Bar in New Zealand where a similar argument for planned colonization has been made partly on the basis of intensive early occupation.

11. lines 662-713 – *I have no idea what I am looking at in these micromorph pictures. For example, in K, I see neither the bone nor the tooth. I wonder if these would work better as separate figures with separate captions, and maybe some arrows to point out particular features. Also N and O seem like the same section but with different imaging. What is the purpose of this? The transition the authors talk about is certainly more visible in O than N.*

This figure has been modified, showing clear scale bars and annotations to specific key features mentioned in the text. The difference between image N and O is that one is in plane polarised light and one in cross polarised light. The latter shows up isotropic minerals more clearly so it is ideal for showing calcium carbonate or objects made of calcium carbonate (e.g. shells, ashes).

Supplement

12. lines 9-11 – *The caption needs better explanation. I do not see cave breccia in the figure. Breccia is in the top left of the figure. We have expanded the caption.*

13. Figure S5 – *I do not see the horizontal combustion features*
We have labelled the combustion features.

14. line 143 – *Why were 90-125 μm grains used instead of the 180-212? While neither size provides single-grain resolution with the 300 μm holes in the disks, much fewer grains will fit into the holes for 180-212. So you actually have small aliquot not single-grain resolution. This is particularly the case given the high acceptance rates of some of the samples.*
The 90-125 μm grain fraction was used as the number of 180-212 μm grains obtained from the samples were too low to be used for measurements. The reviewer has misunderstood, the 300 μm holes were not filled with grains, single 90-125 μm grains were carefully placed in individual holes. We have rephrased this for clarity.

15. lines 178-180 – *I think more justification is needed for applying this criterion given that a large number of grains were rejected for this reason*

We note that this is standard practice in single-grain OSL dating and beyond the scope of this paper. We have rephrased and provided extra citations for this criterion.

16. line 227 – *Not sure what you mean by “reproducible” here*
We agree this is unclear and have replaced it with “consistent”.

17. lines 244-255 – *You do not mention the possibility of beta heterogeneity for the higher over-dispersion in these samples.*

We have rewritten this sentence and added the possibility of beta heterogeneity in the sediment including a new citation to this effect.

18. lines 263-264 – I am not sure about applying nMad for removing outliers. How do you know they do not represent a separate component, as some of the radial graphs seem to suggest?

Even if they do represent a separate component, they are still outliers and should be removed from the main component.

19. lines 269-277 – What value was input for the over-dispersion typical of a single-aged distribution in the finite mixture model? Can you justify using the most abundant component for age determination? Why was the FMM not applied to L19-10A, which had very high over-dispersion?

The FMM was chosen for sample L19-4 as it was the most overdispersed sample and close to the overlying layer, therefore more vulnerable to mixing. When deriving the final FMM D_e value the sigma_b value was incrementally increased, and the sigma_b and component number combination yielding the lowest BIC score (the most statistically optimal) was used. As the sigma_b value was within the range of that obtained for the single-component samples from the site it was considered reliable. The most abundant component produced a near identical age to the single component sample L19-2, also taken from the top of Layer 21. When running the FMM for sample L19-10A the BIC score indicated that the D_e dataset was only characterised by a single D_e component. L19-10A in fact had the lowest overdispersion of the five Layer 21 samples.

Reviewer #3

I very much enjoyed reading this manuscript and I think the analysis of the material from this extremely interesting site and what it represents to the field deserve publication in this prominent journal.

We are very grateful for the endorsement of the reviewer

The site seems well excavated and all the material analysed appropriately. The stratigraphic evidence seems to support the conclusion of this abrupt onset of occupation, both on the macro and microscale.

We thank the reviewer for their positive appraisal of the various aspects our study.

I was asked specifically to look at the radiocarbon dating component of this manuscript. The first thing I noticed that was missing, was the methods section on how the radiocarbon dates were obtained. As elaborately as this is explained in the Supplementary information for the OSL dates, it is as absent for the radiocarbon dates. The chemical pre-treatment of samples when radiocarbon dating close to the limit of the method is very important. For charcoal samples it can make quite a difference when an acid-only, ABA or ABOX treatment is employed (see Bird et al. 2014 for example). The same goes for the shells, how were these treated? Were these shells checked for any diagenetic features? There is for example a density separation technique to remove recrystallised calcite or aragonite (if this is present

of course). This treatment may not always be necessary but it is worth checking whether or not such a treatment would be required.

I think the manuscript really needs to include a section in the supplementary information dedicated solely to radiocarbon dating, which would contain: how the samples were treated, which protocols were performed, the sample sizes and the quality criteria that were followed. Also the error values on the F14C/pmc need to be reported. Ideally also the blank/background values that were used for the blank/background correction of samples, the $\delta^{13}\text{C}$ AMS values and the produced current would be reported but this is not essential.

We have added a new section to the Supplementary Information (Supplementary Note 1) detailing radiocarbon sampling and pretreatment methods. We have also added to the Supplementary Data the radiocarbon ages and errors, the error values for the F14C, and, where available, the $\delta^{13}\text{C}$ values.

In line 354 of the supplementary file: a possible temporal gab... should this be gap?

Yes, typo corrected.

I was wondering which radiocarbon dates were performed as part of this work and which ones were published in Hawkins et al. 2017? Could this be indicated this in the excel file?

We have now indicated this in the Supplementary Data with new columns denoting the year the samples were collected and where they are originally published.

The way the Oxcal model (phases and sequences) is constructed seems fair. However, for the sake of completeness, the entire model's code should also be reported in the supplementary information. I understand it takes up a lot of space but it needs to be included.

We have now added this code to Supplementary Note 3.

I have some other comments, please see in addition to these some comments in the pdf file, which are very minor and can be taken as suggestions.

Q: I was wondering: why it is so unusual, to have sedimentation without human occupation? Isn't this normal natural deposition of sediment, even in caves and rock shelters?

It is not normal in Wallacea. Our hypothesis (published in Louys et al 2017) being this is due to the lack of non-human terrestrial large animal occupants that introduce organic sediment which then traps minerogenic sediment. We have rephrased this in the introduction, adding an additional sentence (lines 60-63).

Q: I find the contrast in Supplementary Figure 4 between the two layers not that clear. In the photo on the left layer 21 is brown-reddish and layer 20 white/grey, while this seems the opposite in the photo on the right side. From the text, layer 21 should be dark organic rich, so I don't fully understand how the photograph on the right side is oriented. The photos appear a bit over lighted actually. This may seem unfair because I know how tricky it can be to take photographs in cave environments.

Yes, it was difficult to capture these images effectively in the low light at the base of the trench. But the reviewer may be mistaken as Layer 21 is the archaeologically sterile layer (hence it looks white where lit and a slightly darker orangey brown where in shadow) and Layer 20 is the dark occupation horizon. We have reduced the contrast in these images and moved the labels around to facilitate their interpretation. We also now specify in the

caption that Layer 21 is exposed horizontally during excavation while Layer 20 is visible vertically in the section.

Q: Fig 3. I'm missing a scale in all these images in Figure 3 actually, could this be added?
Scale bars have been added to the microphotos and dimensions of the thin sections have been included in the caption.

Q: *Has any Scanning Electron Microscopy been done to identify the nature of the red staining and verify that it is indeed iron?*

No, but all macroscopic and microscopic observations are consistent with iron oxide as the source of the colour.

Finally, I am not a micromorphologist so my comments and questions on this may seem odd or obvious but I think it would be helpful for readers to perhaps get a bit more guidance on what we are looking at. Would it possible to circle or otherwise highlight/indicate some of the features (bone, charcoal, shells, debitage) in these figures? I understand this may make it very cluttered so perhaps for a few images. Or alternatively, it might be possible to add a copy of this image in which features are highlighted or circled? This would allow people with advanced understanding of soil micromorphology to look at the figures unimpededly and be helpful to others to find the features they are looking for more easily. Just a thought, simply because I personally would find this very interesting.

We thank the reviewer and agree that the images may be a little daunting without sufficient annotation. We have added annotations so that key characteristics can easily be located in the microphotos.

Reviewer #4 (Remarks to the Author):

This is a well put together paper describing the archaeology at Laili, a highly significant site in Timor-Leste, the dating of which has significant implications for the settlement of the area. The timing of human arrival into Sahul is something that has been hotly contested in recent times, so this new information from Wallacea is of great importance. A clear signature of human habitation beginning ~44ka is consistent with dates from the very few other early sites in Timor Leste reported thus far.

The conclusions are well-supported by the archaeological data which is presented, including secure archaeological dates and well-described differences in the material and archaeozoological deposits in the various chronological layers of the site, clearly showing evidence for changing human use of the site over time (and lack there-of in the case of natural deposits). The site described is rare in the region, with a record going back to first human arrival (not many cave sites in Timor Leste), so reporting of this site and its dates is of great importance. Standard archaeological conventions have been followed. The analyses and presentation of this data are up to standard.

We thank the reviewer for their very positive comments on our manuscript.

Reviewers' Comments:

Reviewer #1:

Remarks to the Author:

I think the authors addressed the previous questions and concerns. Just minor few comments/suggestions:

1. Would it be possible to provide citations/footnotes on the classifications used on lithic analysis in the supplementary information? Additional notes on the protocol.
2. More on use-wear. Would it be possible to add technical specifications of the microscopes that were used? For metallographic microscope – objectives, numerical aperture, working distance. For stereo microscope – optical range, increments. Also, mention the cameras and software that were used in capturing the images as well adapters (if there is any). How were the tools mounted? Handling of the sample during the analysis with stereo and reflected light microscopes? Maybe just additional info about the protocol but I think that will be for the PhD use-wear research. Looking forward to the use-wear paper on this assemblage.
3. For the cleaning method of microscopic analysis, was alcohol used to further clean the edges prior to initial analysis under the reflected light microscope? Especially with Fig. 7A and 7B.
4. Could you provide further discussion or a few lines on hammerstones in relation to the discussion on lithic technology? One of the main arguments of abrupt and intense human occupation is on the production of lithics and that would help the readers understand the processes/methods/techniques behind the technology(ies) mentioned in the text. Also, discuss this percussion technique in relation to bipolar technology, tool miniaturization, production of notched tools, and readily available chert raw materials near the site. Would be interesting to give an overview of this aspect of prehistoric technology.
5. Other comments/ suggestions. Provide labels for all the scale bars in the figures instead of just a bar. Some do not have "1cm". Just make it standardized. Please clean/ erase some remnants of the original background in Fig. 5 (bottom left image, refitted image).

I recommend this for publication once the revision comments of all the reviewers are addressed.

Reviewer #2:

Remarks to the Author:

This is an improved version of this paper, but I still have a problem with the characterization of the occupation as planned mass migration. The authors have presented convincing evidence of a migration, but I am not sure how intense it was. The initial occupation at Laili seemed more intense than what came later at the site and what seems apparent at earlier sites and even at contemporary sites in the region, but the only parallel in initial intensity the authors draw is with a later migration to New Zealand. Thus they have a sample of one and it is hard to gauge how many people were actually involved. A few people living in one place for an extended time can build up a dense occupation litter. The only evidence they present for the migration being planned is the use of rafts instead of canoes, although it is not clear to me what the link is between rafts and planning. At any rate, I think motivation is extremely difficult if not impossible to assess from the archaeological record. How can you get inside the head of someone who lived 50,000 years ago? It is hard enough understanding the motivations of contemporary people or even of one's self. Prosecuting attorneys rely on convincing juries, not science to establish motives. Archaeologists can establish what people did, not what they thought about it. I think the authors should tone down their conclusions by focusing on what they do have evidence for: a migration but not necessarily a massive or planned one.

My only specific comments are on the supplemental luminescence section – my area of expertise.
Supplement

Line 181 – You say in your response to my previous comment that 90-125 micron grains were used because there was insufficient amount of 180-212 micron grains. I think you should say that here.

Line 207 – You say in response to my previous comment that the grains were individually placed in each hole. You measured 16,900 grains. That is a tremendous amount of work. This is not the normal way of loading single-grain disks, so I think you should expand on this more, especially how long it took to load one disk and how exactly it was done. Otherwise, readers with knowledge of luminescence might think that it really wasn't done this way, explaining why the upper sediments (where quartz is more sensitive) has lower over-dispersion than lower sediments.

Table 7 – The table is divided into two sections based on color. Samples in the orange-brown section have much higher acceptance rates than those in the beige section. This implies that the source of the quartz differs between the two sections. This requires comment. What does it say about the site formation process? One possible reason is that the samples from the occupation layers were heated, which would increase their sensitivity.

Line 236 – Given the different origins of the quartz between the two sections, dose recovery should have been done on one sample from each section (sensitive and non-sensitive quartz), instead of just one.

Lines 291-293 – Do the secondary components give ages that would make sense if the interpretation of post-depositional mixing is correct? One would expect these ages to be similar to layers immediately above.

Reviewer #3:

Remarks to the Author:

My previous comments have been addressed and I approve of the current version of the manuscript and supplementary files.

Response to reviewers: reviewer comments in italics, our response in bold

Reviewer #1 (Remarks to the Author):

I think the authors addressed the previous questions and concerns.

We thank the reviewer for recognizing that we addressed the previous round of comments.

Just minor few comments/suggestions:

1. Would it be possible to provide citations/footnotes on the classifications used on lithic analysis in the supplementary information? Additional notes on the protocol.

We have now added an extra sentence to the stone artefact analysis method (lines 569-570) and three references which provide details on lithic analysis: one is a standard lithics textbook for work in Sahul, one is a paper on Timor-Leste which gives a full glossary of definitions for lithics terms, and one is a paper on Timor-Leste by the same lithic analyst as the present paper using the same protocols as this work.

2. More on use-wear. Would it be possible to add technical specifications of the microscopes that were used? For metallographic microscope – objectives, numerical aperture, working distance. For stereo microscope – optical range, increments. Also, mention the cameras and software that were used in capturing the images as well adapters (if there is any). How were the tools mounted? Handling of the sample during the analysis with stereo and reflected light microscopes? Maybe just additional info about the protocol but I think that will be for the PhD use-wear research. Looking forward to the use-wear paper on this assemblage.

We have now added the technical specifications for the microscopes, cameras, and software on lines 591-596. We have added a sentence describing mounting detail on lines 597-598, and a sentence describing artefact handling on lines 589-590.

3. For the cleaning method of microscopic analysis, was alcohol used to further clean the edges prior to initial analysis under the reflected light microscope? Especially with Fig. 7A and 7B.

We have added a sentence regarding alcohol cleaning on lines 588-589.

4. Could you provide further discussion or a few lines on hammerstones in relation to the discussion on lithic technology? One of the main arguments of abrupt and intense human occupation is on the production of lithics and that would help the readers understand the

processes/methods/techniques behind the technology(ies) mentioned in the text. Also, discuss this percussion technique in relation to bipolar technology, tool miniaturization, production of notched tools, and readily available chert raw materials near the site. Would be interesting to give an overview of this aspect of prehistoric technology.

We have added an extra sentence discussing hammerstones to the main text (lines 294-297). The range of knapping methods are discussed in the remainder of this paragraph (lines 275-294) with details of bipolar technology, core-on-flakes, discoidal cores, multi- and uni-facial reduction, as well as overhang removal as a platform preparation technique. Tool miniaturization is discussed on lines 298-302, while notched tools are discussed on lines 303-306. The availability of chert is discussed on lines 273-274 and lines 389-393.

5. Other comments/ suggestions. Provide labels for all the scale bars in the figures instead of just a bar. Some do not have "1cm". Just make it standardized. Please clean/ erase some remnants of the original background in Fig. 5 (bottom left image, refitted image).

Scale labels have been added for the in text figures. We are grateful to the reviewer for spotting remnants of the original background in Figure 5 and have now removed this.

I recommend this for publication once the revision comments of all the reviewers are addressed.

We are pleased the reviewer recommends this paper for publication.

Reviewer #2 (Remarks to the Author):

This is an improved version of this paper, but I still have a problem with the characterization of the occupation as planned mass migration. The authors have presented convincing evidence of a migration, but I am not sure how intense it was. The initial occupation at Laili seemed more intense than what came later at the site and what seems apparent at earlier sites and even at contemporary sites in the region, but the only parallel in initial intensity the authors draw is with a later migration to New Zealand. Thus they have a sample of one and it is hard to gauge how many people were actually involved. A few people living in one place for an extended time can build up a dense occupation litter. The only evidence they present for the migration being planned is the use of rafts instead of canoes, although it is not clear to me what the link is between rafts and planning. At any rate, I think motivation is extremely difficult if not impossible to assess from the archaeological record. How can you get inside the head of someone who lived 50,000 years ago? It is hard enough understanding the motivations of contemporary people or even of one's self. Prosecuting attorneys rely on convincing juries, not science to establish motives. Archaeologists can establish what people did, not what they thought about it. I think the authors should tone down their conclusions by focusing on what they do have evidence for: a migration but not necessarily a massive or planned one.

In accordance with the reviewer's previous recommendation we removed all reference to planning from the last version of the manuscript. We replaced 'planned' in one instance in the final paragraph with 'purposive', a significantly lower bar than planning as it only implies that migration was done with the purpose of migrating, rather than with a multi-stage series of intermediary goals. The distinction between deliberate migration and being accidentally swept out to sea and washed up on an island, is widespread in archaeological literature on island colonisation and considered a primary goal of island archaeology. We have added a new citation to underscore the distinction we are making ¹. We have removed the reference to Wairau Bar that the reviewer objects to on the grounds of it being a single site, but note that by their very nature such initial colonisation signatures will be rare. We agree with the reviewer that a few people living in one place for an extended period of time can build-up a dense occupation litter, but Laili allows us to test between that and the alternative scenario of intensive occupation resulting in a dense occupation litter, as it has relatively rapid non-anthropogenic sediment accumulation. We now articulate this alternative scenario of a gradually increasing anthropogenic signature against the background of non-anthropogenic sedimentation on lines 405-408. We make no claims for planning in relation to the use of rafts over canoes. We do not speculate on the motivations for colonisation, our argument is merely that it was not accidental. We recognize that 'mass-migration' is a well-worn phrase that might conjure the wrong image of wholesale population movement, therefore we have replaced this with 'large-scale' to distinguish between the evidence from Laili and, for example, the Lower Palaeolithic occupation of Flores where the earliest thus far known site, Wolo Sege, has just 48 stone artefacts excavated over a larger area ². Our argument that the colonization was purposive is also supported by the aquatic foraging adaptation that characterizes the early occupation but contrasts with the rest of the sequence (lines 421-429 and 476-479). We now make it clear in the conclusion that our argument for purposive colonization is relative to the necessarily small founding population of passive dispersal (lines 479-481).

My only specific comments are on the supplemental luminescence section – my area of expertise.

Supplement

Line 181 – You say in your response to my previous comment that 90-125 micron grains were used because there was insufficient amount of 180-212 micron grains. I think you should say that here.

We have added a sentence to clarify this (lines 184-185).

Line 207 – You say in response to my previous comment that the grains were individually placed in each hole. You measured 16,900 grains. That is a tremendous amount of work. This is not the normal way of loading single-grain disks, so I think you should expand on this more, especially how long it took to load one disk and how exactly it was done. Otherwise, readers with knowledge of luminescence might think that it really wasn't done this way, explaining why the upper sediments (where quartz is more sensitive) has lower over-dispersion than lower sediments.

The reviewer is correct, this was indeed a tremendous amount of work and took many months. We took the time to ensure that discs were loaded precisely. Standard methods for the loading of single-grain discs were employed, with grains moved carefully across the disc surface and into each hole under magnification using a fine brush trimmed back to a few hairs. Any extra grains were removed with the point of a pin that had been rubbed to produce static electricity. We now specify this loading protocol on lines 190-193, and that 16,900 grains represents many months work on line 215.

Table 7 – The table is divided into two sections based on color. Samples in the orange-brown section have much higher acceptance rates than those in the beige section. This implies that the source of the quartz differs between the two sections. This requires comment. What does it say about the site formation process? One possible reason is that the samples from the occupation layers were heated, which would increase their sensitivity.

The reviewer is correct, the samples from the occupation layers almost certainly experienced heating as these layers contain very high quantities of combustion signatures (detailed in both the manuscript and supplementary information), and this is the most likely explanation for the difference in sensitivity — and hence the acceptance rates of grains — between samples from the sterile and the occupation layers. We have added this explanation to the supplementary information on lines 235-238.

Line 236 – Given the different origins of the quartz between the two sections, dose recovery should have been done on one sample from each section (sensitive and non-sensitive quartz), instead of just one.

No, as the reviewer says in the previous comment there is a ready explanation for the difference in sensitivity between the quartz from the different layers, it is not due to their differing origins, it is due to one group being post-depositionally heated. There is no evidence from our geoarchaeological analyses to suggest a change in sedimentation regime that would mean a different source for the quartz grains.

Lines 291-293 – Do the secondary components give ages that would make sense if the interpretation of post-depositional mixing is correct? One would expect these ages to be similar to layers immediately above.

Optical dating of individually placed quartz grains can estimate the true burial age of a grain. The range of post-depositional processes (burrowing of insects and small animals, plant root growth etc.) that can produce mixing of sediments can cause grains to move downward into an underlying deposit, as the reviewer states, but also upwards into overlying sediments. Samples from both the sterile and occupation layers at Laili demonstrate both upward and downward movement of a small number of grains, and thus do not show a pattern of the intrusive second/third component grains deriving only from the layer immediately above the sample locations. As such we would not expect the

second/third component age estimates to necessarily correspond to the deposition age of the layer directly above a given sample, but rather show a spectrum of younger and older burial ages (compared to the primary population of grains), as indeed we find at Laili.

Reviewer #3 (Remarks to the Author):

My previous comments have been addressed and I approve of the current version of the manuscript and supplementary files.

We are pleased the reviewer approves of the revised version of both the manuscript and supplementary information.

References

- 1 Leppard, T. P. Passive dispersal versus strategic dispersal in island colonization by hominins. *Current Anthropology* 56, 590-595 (2015).**
- 2 Brumm, A. *et al.* Hominins on Flores, Indonesia, by one million years ago. *Nature* 464, 748-752 (2010).**

Reviewers' Comments:

Reviewer #2:

Remarks to the Author:

I think the authors adequately addressed my concerns, although I think there is a fundamental difference between them and me about what archaeologists can say about intention. My view is that is beyond the scope of archaeology, and is not very interesting anyway. What is interesting is what people actually did and why they did it can be addressed without knowing what people thought about it. In fact, focusing on intention begs the question of why something was done. "It was done because they wanted to do it" That is not very illuminating. Replacing "planning" with "purposeful" doesn't really help. (I note that "planning" is still present in line 395.) One could remove all mention of purposeful, and the paper would still be good. The authors demonstrate a large scale migration. Why it happened is still unknown, and saying the people meant to do it does not add anything. However, I think this is a larger archaeological issue than can be dealt with in this paper, so I do not want to hold up this paper that otherwise contains useful data.

I make two other small comments that I think might relate to typos. On line 157 they mention "finger-grained" sedimentation. I do not know what that it and I wonder if they meant "fine-grained". In lines 243-44 in the supplement, I think the authors must mean that acceptance rates were higher for the occupation layers, not lower. That is what the table shows.

Response to Review, reviewer comments in italics our response in bold

I think the authors adequately addressed my concerns, although I think there is a fundamental difference between them and me about what archaeologists can say about intention. My view is that is beyond the scope of archaeology, and is not very interesting anyway. What is interesting is what people actually did and why they did it can be addressed without knowing what people thought about it. In fact, focusing on intention begs the question of why something was done. "It was done because they wanted to do it" That is not very illuminating. Replacing "planning" with "purposeful" doesn't really help. (I note that "planning" is still present in line 395.) One could remove all mention of purposeful, and the paper would still be good. The authors demonstrate a large scale migration. Why it happened is still unknown, and saying the people meant to do it does not add anything. However, I think this is a larger archaeological issue than can be dealt with in this paper, so I do not want to hold up this paper that otherwise contains useful data.

We agree that this a larger issue beyond the scope of this paper and one in which there is divergence between European, particularly Francophone, chaine operatoire literature where understanding intention is an explicit goal, and non-European, Anglophone literature where intention is eschewed. We have removed mention of 'purpose' in line with the editor's comments, however differentiating accidental and strategic island colonisation is a stated aim for many island archaeologists as the Leppard citation demonstrates. We thank the reviewer for noting the word 'planning' on line 395. We have not removed this it as it does not refer to our own work, it refers to a paper published in another *Nature* journal where in the title of the paper they say that colonisation was not accidental, and in the abstract they explicitly make a claim for planning.

I make two other small comments that I think might relate to typos. On line 157 they mention "finger-grained" sedimentation. I do not know what that it and I wonder if they meant "fine-grained".

We have corrected this typo.

In lines 243-44 in the supplement, I think the authors must mean that acceptance rates were higher for the occupation layers, not lower. That is what the table shows.

We have corrected this typo.